

# Warm temperatures, cool sponges: the effect of increased temperatures on the Antarctic sponge *Isodictya* sp.

Marcelo González-Aravena[1,*], Nathan J. Kenny[2,3,*], Magdalena Osorio[1], Alejandro Font[1], Ana Riesgo[2] and César A. Cárdenas[1]

[1] Departamento Científico, Instituto Antártico Chileno, Puntas Arenas, Chile
[2] Life Sciences, The Natural History Museum, London, London, UK
[3] Life Sciences, Oxford Brookes University, Oxford, UK
* These authors contributed equally to this work.

Corresponding authors
Nathan J. Kenny,
nathanjameskenny@gmail.com
César A. Cárdenas,
ccardenas@inach.cl

## ABSTRACT

Although the cellular and molecular responses to exposure to relatively high temperatures (acute thermal stress or heat shock) have been studied previously, only sparse empirical evidence of how it affects cold-water species is available. As climate change becomes more pronounced in areas such as the Western Antarctic Peninsula, both long-term and occasional acute temperature rises will impact species found there, and it has become crucial to understand the capacity of these species to respond to such thermal stress. Here, we use the Antarctic sponge *Isodictya* sp. to investigate how sessile organisms (particularly Porifera) can adjust to acute short-term heat stress, by exposing this species to 3 and 5 °C for 4 h, corresponding to predicted temperatures under high-end 2080 IPCC-SRES scenarios. Assembling a de novo reference transcriptome (90,188 contigs, >93.7% metazoan BUSCO genes) we have begun to discern the molecular response employed by Isodictya to adjust to heat exposure. Our initial analyses suggest that TGF-β, ubiquitin and hedgehog cascades are involved, alongside other genes. However, the degree and type of response changed little from 3 to 5 °C in the time frame examined, suggesting that even moderate rises in temperature could cause stress at the limits of this organism's capacity. Given the importance of sponges to Antarctic ecosystems, our findings are vital for discerning the consequences of short-term increases in Antarctic ocean temperature on these and other species.

## BACKGROUND

Sponges are vital components of a variety of ecosystems in terms of abundance and ecosystem services (*Bell, 2008*; *De Goeij et al., 2013*). This is especially true in the Antarctic, where they are among the most common benthic species, providing habitat and food for a wide range of organisms (*McClintock et al., 2005*; *Cárdenas et al., 2016*; *Cárdenas & Montiel, 2017*; *Gutt et al., 2017*). Sponges, like most marine species, will be broadly, and perhaps adversely, affected by climate change, but at present studies into their capacity to

cope with increases in temperature are limited in scope and number (*Bell et al., 2015*; *Carballo & Bell, 2017*).

The Western Antarctic Peninsula (WAP) is one of the areas of the planet which has experienced some of the most significant changes in air and water temperature (*Turner et al., 2016*; *Stenni et al., 2017*). Current estimates of climate change in the Antarctic suggest that the mean temperature of seawater could rise by about 0.6 °C (Representative Concentration Pathway 2.6) to 2.0 °C (Representative Concentration Pathway 8.5) (*IPCC, 2014*). This is expected to have major implications for Antarctic organisms as they are highly sensitive to environmental variation (*Peck, 2018*; *Ingels et al., 2012*). For this reason the projected changes in water temperature constitutes a major threat to ecosystem function in these waters. Such changes would modify the prevalence of key functional species, thus affecting associated ecosystem processes (*Somero, 2010*). However, we are only beginning to understand the resilience of many species found in the Antarctic to rising temperature conditions (*Peck, 2018*; *Peck et al., 2014*; *Suckling et al., 2015*; *Clark et al., 2017*), and the point to which they will be affected by any changes in mean temperature, for either long or short periods of time. The deleterious effects of temperature exposure in sponges from other latitudes include bleaching of symbionts, tissue necrosis and death (*Ramsby et al., 2018*). It is vital to gain this information in Antarctic sponge species, so that policy decisions can be made with a full understanding of the likely impacts of these changes.

Some sponge species have been suggested to be relatively robust to moderate changes in temperature. Caribbean and Brazilian coral-reef sponges have been investigated and found to survive fluctuations in ambient temperature (*Duckworth et al., 2012*; *Kelmo, Bell & Attrill, 2013*). A recent study experimentally demonstrated that the boreal deep-sea sponge *Geodia barretti* is able to cope with temperature rises with few ill effects (*Strand et al., 2017*). Other sponge species are however not so resilient (*Ramsby et al., 2018*; *Cebrian et al., 2011*; *Webster et al., 2013*), and both reproduction (*Ettinger-Epstein et al., 2007*) and filtration (*Massaro et al., 2012*) in sponges has been shown to be affected by increases in temperature, even over short periods of time. In some cases, this has led to widespread mortality (*Cebrian et al., 2011*; *Cerrano et al., 2000*).

We have little current understanding as to what makes some species able to cope with broad temperature ranges, and it is possible that Antarctic-dwelling species may be particularly vulnerable due to their specialisation for extremely cold, relatively stable temperature conditions (*Peck, 2005*). Their molecular components in particular may have altered over evolutionary time, as has been observed previously in a variety of species (*Clark et al., 2017*), and warmer temperatures could prove deleterious. The sponge *Isodictya* sp. (Fig. 1A) is no exception to this. While this sponge is commonly observed around the WAP and generally lives in waters with temperatures between −1.8 °C (*Klinck et al., 2004*), it is rarely found in warmer areas, with summer peaks around the WAP reaching 1.5 and 2 °C around Marguerite Bay and Palmer Archipelago respectively. Due to its ubiquity and narrow normal temperature range, this sponge is an ideal model to test the consequences of temperature exposure on cold-adapted species. Increasingly this sponge is exposed to warmer summer temperatures across its natural range, with shallow-water

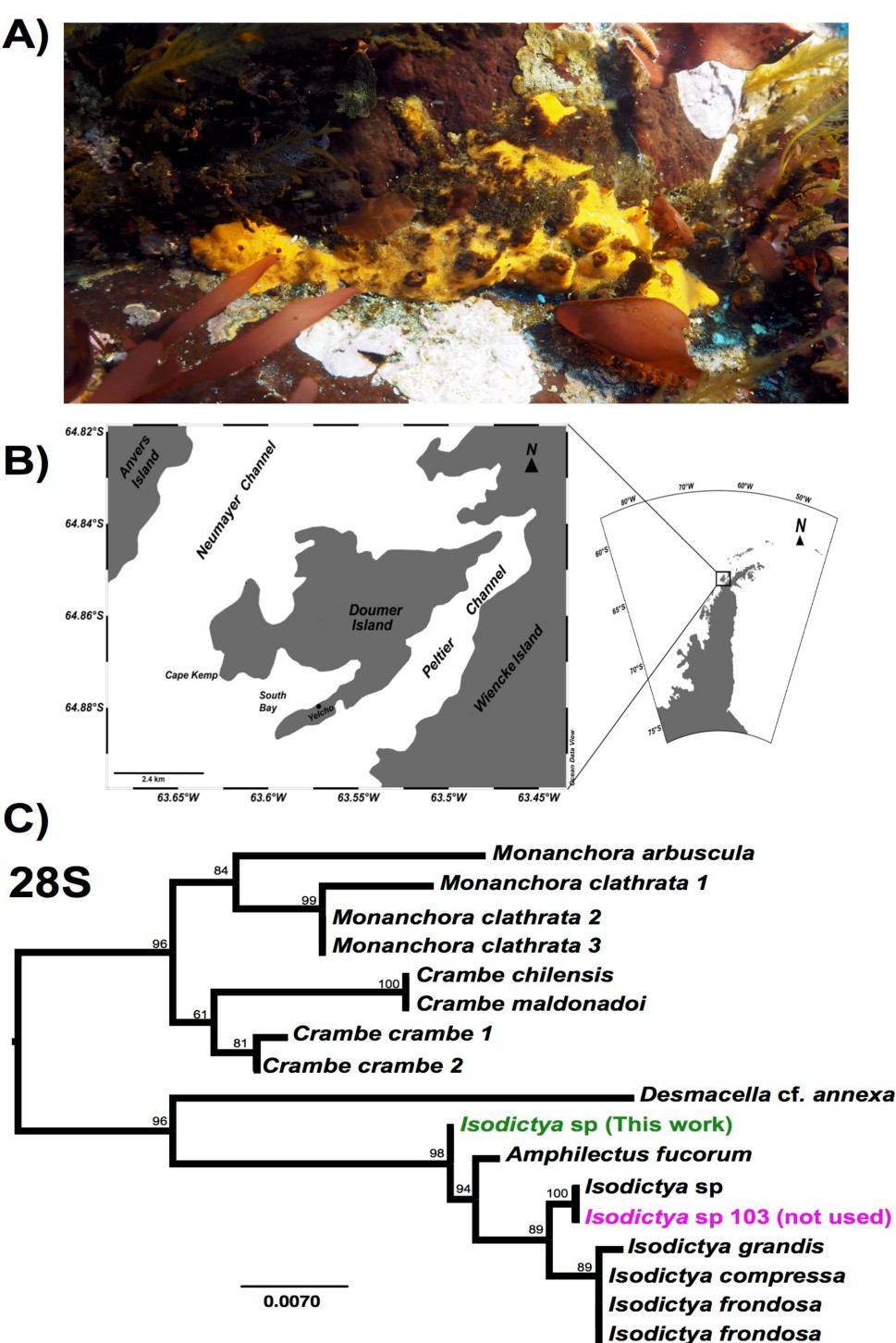

**Figure 1  Image of *Isodictya*, sample site and phylogeny.** (A) Image of *Isodictya* sp. collected at South Bay, WAP (10 m depth). (B) Sample collection location, Doumer Island, WAP. (C) RAxML trees of 28S rRNA under the GTR model from the *Isodictya* species mentioned in this manuscript, along with those of related species. Note the difference in placement of the samples used in the present manuscript (in green, *Isodictya* sp., this work) to that of one of the samples sequenced for this work, *Isodictya* sp. 103 (in magenta), which was excluded due to it being identified as a cryptic, but separate, species.

temperatures reaching 3 °C in some places around the WAP (*Cárdenas, González-Aravena & Santibañez, 2018*).

Illumina-based sequencing can reveal both subtle and broad changes in expression in studies of model and non-model organisms (*Cahais et al., 2012*). These studies have the additional benefit of also providing a suite of information on the gene sequences of the organism in question, which is particularly useful for non-model species. While genomic-level datasets within the Porifera remain depauperate and limit our understanding of adaptation to particular conditions, a number of investigations into the transcriptomic complements of a variety of sponges have now been successfully performed (*Pérez-Porro et al., 2013*; *Riesgo et al., 2014*; *Fernandez-Valverde, Calcino & Degnan, 2015*). The transcriptional changes in response to temperature shifts that occur within sponges themselves (*Webster et al., 2013*; *Guzman & Conaco, 2016*) and changes in their bacterial symbionts (*Ramsby et al., 2018*; *Erwin et al., 2012*; *Fan et al., 2013*; *Pita et al., 2013*; *Simister et al., 2012*) have also only recently begun to be unveiled. Some changes in gene expression revealed by these studies, such as those in HSP70 expression (*Guzman & Conaco, 2016*; *Lopez-Legentil et al., 2008*) are easily explained with reference to known metazoan responses to temperature fluctuation and heat shock. Similarly, responses such as the up-regulation of genes involved in signal transduction, tissue morphogenesis, cell signalling and antioxidant activity are all of clear utility to tissues as they adapt to thermal stresses. What has yet to be tested thoroughly is whether these changes are mirrored in extremophile species. It is possible that gene complements adapted to extremes may be less able to cope with heat shock, particularly over rapid time scales.

Our study demonstrates that some extremophile species such as *Isodictya* sp. may be able to respond to acute short term stress, but that there may be limits to the upper bounds of this response. Using a small number of replicates and two 'heat shock' regimes, we have been able to use these findings to investigate the specific pathways used by these organisms, and contrast these with previous findings in species adapted to warmer temperature regimes. The sponge studied here, *Isodictya* sp. is one of the most common species in shallow-water rocky reefs (<25 m depth) around Doumer Island, Palmer Archipelago, WAP (Fig. 1B).

In this manuscript we present the first transcriptomic analysis of the effect of acute warming on a cold-adapted sponge. Due to the constraints of working with polar species in Antarctic conditions the numbers of replicates is low, particularly for the control specimen. This limits the strength of some of the conclusions that can be drawn from this work. Nevertheless, given its novelty this knowledge will be vital for understanding the impact of temperature rises on Antarctic species, but also for discrete, targeted investigation of the role of specific pathways in temperature adaptation in the future.

## METHODS

### Animal collection and heat treatment

Adult samples of six individuals of the Antarctic sponge *Isodictya* sp. were collected by SCUBA diving at 10 m depth in Cape Kemp, off Yelcho Research Station, Doumer Island, WAP (64°51′S 63°35′W, Fig. 1B) in January 2017. The study was conducted under permit

806/2015 granted by the Chilean Antarctic Institute (INACH). Sponges were then transferred to the laboratory at Yelcho Station, where they were maintained in 140 L fibreglass tanks with unfiltered, flowing seawater (approximately 0.5 °C, pumped directly from the sea floor near to collection site (5–10 m depth)) for a week to allow acclimation to laboratory conditions. Tanks were covered with two layers of one mm shade cloth (fiberglass 50% neutral density screen) to represent light levels occurring in situ. Relative light intensity levels and temperature in the experimental tanks were measured (one measurement every 15 min) during the duration of experiment with a HOBO Pendant® temperature/light data logger (Onset, USA). Sponges were then placed into one of three possible treatment conditions—a control tank, where samples were maintained in seawater pumped directly from the sea floor at approximately 0.5 °C, or one of two possible treatment temperatures, 3 and 5 °C, with water warmed in a header tank before pumping into the treatment area, and subsequently maintained at a set temperature using SOBO Aquarium heaters (500 watt). These temperatures were chosen in line with empirical evidence of warming in the WAP (*Cárdenas, González-Aravena & Santibañez, 2018*). Sponge samples were placed in individual treatment tanks, for a total of three treatments and two replicates (2 × control, 2 × 3 °C, 2 × 5 °C) although one control sample was not used for analysis as described in 'Results and Discussion'.

## RNA extraction and sequencing

Sponges were taken from tanks after 4 h of exposure to the control/treatment conditions, and a single fragment of each sample, approx. one $cm^3$, was taken immediately and placed in RNAlater (Qiagen, Hilden, Germany) and stored at −20 °C on site, before shipment on dry ice (with an unbroken cold chain) from Yelcho via Escudero Station to a molecular laboratory at the Laboratorio de Biorrecursos at INACH, Punta Arenas. RNA was extracted from sponge samples using E.Z.N.A Total RNA kit II (Omega Bio-Tek Inc., Norcross, GA, USA) according to the manufacturer's protocol with a previous step of homogenisation with liquid nitrogen. RNA quality and quantity was determined on a 2100 Bioanalyser (Agilent Technologies, Santa Clara, CA, USA). Quality control obtained an average RIN value of 8.2. The sole sample with a RIN value less than 7 (a control sample, with RIN 6.2) was not included in this work. One μg of RNA from each sample was used to produce RNA libraries for sequencing, with TruSeq Stranded mRNA kit (Illumina, San Diego, CA, USA), 100 bp nominal intra-read fragment size, 15 rounds of amplification and adaptors AGTTCC, ATGTCA, CCGTCC, GTCCGC and CGATGT.

Libraries were sequenced on a Hiseq2500 sequencer by the Macrogen provider, using approximately one half of a run in total. The sequencing provider performed initial assessment of read quality and de-multiplexing of reads according to their procedures, and provided us with paired-end reads for download from an external server, with no unpaired orphan reads retained by this process.

## Quality control and assembly

We confirmed the removal of adapter sequences and overall sequence quality using the FastQC programme (*Andrews, 2010*). Low-quality regions of reads were trimmed using

Trimmomatic 0.33 (*Bolger, Lohse & Usadel, 2014*) with the following settings: ILLUMINACLIP:../Adaptors.fa:2:30:10 LEADING:3 TRAILING:3 SLIDINGWINDOW: 4:20 MINLEN:30 where the Adaptors.fa file consisted of the appropriate indexes for the libraries in question. The resulting trimmed reads were then compressed using gzip and re-analysed with FastQC. Seqtk fqchk (*Li, 2013*) was used to determine average sequence quality scores.

Trimmed reads from all five samples were assembled into a reference transcriptome using Trinity version 2013_08_14 (*Grabherr et al., 2011*), with two non-standard settings: a minimum contig length of 200 bp and in silico read normalisation. Clustering of isoforms was performed natively by Trinity, but the assembly was not 'flattened' to preserve isoformal and splice information for future work. DeconSeq standalone version 0.4.3 (*Schmieder & Edwards, 2011*) was used to remove contamination, with settings -i 50 -c 50, using pre-prepared bacterial, archean, hsref and viral databases (2,206, 155, 1 and 3,761 complete genomes respectively), resulting in our final reference assembly. This is available from the DOI and URL below as File S1.

## Phylogenetic analysis

28S and HSP70 sequences were extracted from our assemblies using BLASTN (*Altschul et al., 1990*) on a local server, using sequences of known orthology from Genbank as search queries. These were aligned to previously published sequences downloaded from the NCBI database using MAFFT (*Katoh & Standley, 2013*). HSP70 gene alignments were cured using Gblocks (*Castresana, 2000*), although this was not necessary for the well-conserved 28S sequence. Phylogenetic analysis was run in RAxML GUI (*Silvestro & Michalak, 2012*) using ML + Rapid Bootstrap, the GTR model (28S)/LG+I+G model (HSP70) and 1,000 bootstrap replicates. Phylogenetic and molecular evolutionary analyses of AIF genes were conducted using MEGA version 7.0 (*Kumar, Stecher & Tamura, 2016*). The tree was inferred using the Neighbor-Joining method, based on the alignment of the sequences using ClustalW 2.0 (*Larkin et al., 2007*) (alignment was improved using the Seaview software 4.6.5 (*Gouy, Guindon & Gascuel, 2009*)). Resultant tree topologies were evaluated by bootstrap analyses based on 1,000 replicates.

## Transcriptome analysis and annotation

To assess the content of the transcriptome Basic Universal Single Copy Orthologue (BUSCO) v1.1b1 (*Simao et al., 2015*) was run against the reference transcriptome, using the eukaryotic and metazoan BUSCO gene lists to estimate completeness. Annotation of contigs was performed by translating the longest ORF for each contig using the getORF. py python script, taking the longest ORF, then using standalone BLASTP (*Altschul et al., 1990*) (cutoff *E*-value 0.000001) to search against the refseq protein database. This was supplemented using Blast2GO Pro (*Conesa et al., 2005*), where full InterPro scanning, mapping, annotation (including ANNEX) and enzyme code mapping was performed, the complete annotations for which are available as File S2.

The KAAS-KEGG automatic annotation server was used to gain an understanding of the recovery of complete pathways in our transcriptome. These were generated using the

online tool (*Moriya et al., 2007*) rather than as integrated into Blast2GO, due to the increased functionality of the standalone server. The bi-directional best hit method was used to identify and annotate the longest orfs from our contigs, with the protein sequences generated earlier used as the basis for these comparisons against a range of eukaryotic species. The maps generated by this were compared directly against extant ones previously generated for the sponge *Amphimedon queenslandica*, and the complete annotations are available as File S3.

## Differential expression analysis and over-representation analysis

Comparative analyses of gene expression were performed using RSEM (*Li & Dewey, 2011*) as packaged within the Trinity (*Grabherr et al., 2011*) module, using Bowtie2 (*Langmead & Salzberg, 2012*) for alignment. The three treatments (control, 3 and 5 °C) had their replicates (1, 2 and 2 respectively) cross-sample normalised according to Trimmed Mean of $M$-values. The three treatments were then cross-compared using edgeR (*Robinson, McCarthy & Smyth, 2010*) within the Trinity (*Grabherr et al., 2011*) perl wrapper script, according to the 'classic' pairwise model with a *p*-value cut off for FDR of 0.0001, a minimum absolute ($\log2(a/b)$) change of 2 (i.e. fourfold change) and a dispersion setting of 0.1, deliberately over-specified to preclude false positive identification of over-expression. We excluded any differentially expressed contigs where transcription was only detected in a single sample of any of the five used in this analysis, prior to clustering and any further analysis or display, to avoid spurious results caused by transient expression or contamination of single samples. As with the dispersion estimate, the *p*-value cutoff and minimum absolute change values are stringent, to exclude potentially artifactual changes from our analysis, given the small number of replicates. The remaining differentially-expressed gene lists were then targeted for further analysis. These results are provided as File S4.

Gene Ontology (GO) over-representation analysis was performed in Blast2GO Pro (*Conesa et al., 2005*) using Fisher's Exact Test, with FDR control for multiple testing, with a *p*-value cutoff of 0.05 (*Benjamini & Hochberg, 1995*). The test sets were the annotated results for the sequences present in each of the over-represented sets derived from the differential expression analysis, analysed in turn, with the reference set the full list of annotated contigs, minus those in the test set.

# RESULTS AND DISCUSSION

## Sequencing and read cleaning

While five individual samples are noted here (a single control, and two replicates for both the 3 and 5 °C treatments) another control sample was also initially taken and sequencing performed as described above. Despite being morphologically identical, our initial assays of the sequences from that sample, and particularly the sequence of common molecular markers used in phylogenetic comparison, including *28S rRNA* and *cytochrome c oxidase 1* markers, revealed that it is in fact a cryptic related species, which will be the subject of description in the future (see Fig. 1C). We have not included this sample in any

**Table 1 Raw read data, before and after cleaning.**

| Metric | Control (209) | 3 °C Replicate A (117) | 3 °C Replicate B (123) | 5 °C Replicate A (135) | 5 °C Replicate B (141) |
|---|---|---|---|---|---|
| Total read pairs (initial) | 25,988,569 | 28,513,364 | 26,960,296 | 25,793,570 | 26,289,401 |
| Total read pairs (after cleaning) | 18,784,882 | 20,653,864 | 19,553,789 | 18,122,106 | 18,763,103 |
| Total bases in all reads (initial) | 5,249,690,938 | 5,759,699,528 | 5,445,979,792 | 5,210,301,140 | 5,310,459,002 |
| Total bases in all reads (after cleaning) | 3,591,119,834 | 3,940,343,972 | 3,742,325,651 | 3,441,581,533 | 3,575,958,067 |
| GC% (initial) | 45 | 46 | 45 | 46 | 45 |
| GC% (after cleaning) | 45 | 46 | 44 | 46 | 44 |
| Average Q (initial) | 35.75 | 35.75 | 35.8 | 35.55 | 35.75 |
| Average Q (after cleaning) | 37.45 | 37.4 | 37.45 | 37.4 | 37.4 |
| % less than Q20 (initial) | 4.55 | 4.55 | 4.55 | 5.05 | 4.7 |
| % less than Q20 (after cleaning) | 0.2 | 0.2 | 0.2 | 0.25 | 0.25 |

of the analyses in this work. The lack of a second control sample meant that comparisons of the treatment conditions limited some conclusions, as discussed later in this manuscript.

Basic sequencing metrics can be seen in Table 1, alongside those after read cleaning. Our initial FastQC analysis revealed the presence of low quality nucleotide sequence in the second file of many pairs. Cleaning was therefore stringent, and resulted in markedly fewer, but much better average quality, reads for all samples, which were then used for assembly of the reference transcriptome and for differential expression analysis. GC% (which can be a crude proxy for contamination or changes in expression) was even through all our samples, between 44% and 46%, and changed little with cleaning. A small number of over-represented sequences were also initially observed in our reads, as is commonly observed in Illumina-based transcriptomic analysis due to known biases in hexamer binding (*Hansen, Brenner & Dudoit, 2010*). The difference in average quality after cleaning was modest, with an average difference of around two in Phred score. Our original reads have been uploaded to the NCBI SRA with accession number PRJNA415418.

## Assembly and completeness

Reads from all five samples were used to construct a reference assembly (Table 2), for use in further analysis (File S1). We checked this for potential contamination using Deconseq, comparing our data to known bacterial, viral, 'archaea' (sensu (*Woese, Kandler & Wheelis, 1990*), we are aware of archaean paraphyly) and human genomic sequences. With very permissive settings for recognition as foreign sequence (minimum thresholds for removal, 50% similarity across at least 50% of the contig length) 797 contigs were tagged as potential contamination. These were removed from our dataset before further analysis. It should be noted, however, that particularly novel bacterial sequences not yet represented in the *nr* database may still be present in our data, despite the use of polyA sequencing, as these could not be represented in our removal database and therefore will not have been removed. While beyond the scope of this manuscript, changes in symbiont content may be vital for long-term adaptation to change.

| Table 2 Statistics, reference transcriptome assembly. | |
|---|---|
| Number of transcripts | 90,188 |
| Number of trinity 'genes' | 70,844 |
| Total bp in assembly | 59,274,448 |
| Max contig length (bp) | 19,068 |
| Mean contig length (bp) | 657.23 |
| Median contig length (bp) | 338 |
| % GC | 43.10% |
| N20 contig length | 2,921 |
| N50 contig length | 1,113 |
| # contigs in N50 | 12,997 |
| Number of transcripts over 1,000 bp | 14,633 |
| Transcripts w/blast hit | 20,607 |
| Transcripts w/GO term | 12,924 |

Statistics related to our cleaned reference assembly can be seen in Table 2. A total of 90,188 contigs are present, and Trinity has automatically assigned 70,844 as independent 'genes'. A small amount of heterozygosity or potential splice variation has therefore been recognised, with 19,344 contigs flagged as isoformal variants of other contigs in our assembly. This is not unusual, and is in fact less than that recognised in some other de novo assemblies (for instance, (Kenny et al., 2018), where every gene possessed approximately two 'isoform' variants on average). Our samples may therefore possess little in the way of genetic variability at most loci. Our reference assembly is well-assembled, with a high N50 (1,113 bp). 12,997 contigs were longer than this N50 value, and 14,633 longer than 1 kb in length.

To test the completeness of our transcriptomic dataset we used the BUSCO approach (Simao et al., 2015), which revealed our dataset to be remarkably complete. Of 978 BUSCO orthogroups plausibly expected in any metazoan species, 916 complete BUSCOs were found (654 single copy, 262 duplicated, possibly reflecting allelic or isoform differences). A total of 20 were present only as fragmentary sequence, while 42 were missing. Of the 303 eukaryotic BUSCO sequences, 298 (98.4%) were present, 213 as single copy, 85 with duplicates and two as fragmentary sequence. Only three BUSCO groups were missing. By way of comparison, the published draft of the A. queenslandica set is missing 1.6% of the eukaryote set (five genes), and 4.9% (49) of the metazoan complement (Fernandez-Valverde, Calcino & Degnan, 2015).

## Annotation and content

To annotate our data, we used automated methods, including the Blast2GOPro and KEGG platforms. Of 90,188 total contigs, 20,607 were given an initial annotation based on BLAST results vs the Refseq database at an $E$-value cutoff of $10^{-6}$, chosen empirically to allow maximum annotation ability without introducing artefacts. Of these contigs, 7,303 had their 'best hit' to proteins in the A. queenslandica genome. The diatoms Phaeodactylum

*tricornutum CCAP 1055/1* and *Thalassiosira pseudonana CCMP1335* were observed as 'best hit' species for 798 and 563 contigs respectively, which is not surprising as previous work as described abundant presence of diatoms in other Antarctic sponges (*Cerrano et al., 2000*, *2004*). Annotated diatom data was not included in further analysis. No obvious bacterial, viral, human or archaeal contamination was observed, although as described earlier, 797 contigs were removed from our assembly by Deconseq before any further analysis.

More discrete annotation was then performed using Blast2GO Pro. Of the 20,607 contigs preliminarily annotated with Blast, 6,267 could not be annotated further. A total of 1,389 were able to be 'mapped' to GO terms, but were unable to be annotated further. A total of 12,924 contigs received GO annotations (File S2). A total of 27 additional contigs had Interproscan results and were also placed into GO categories. While the expected total gene complement of a metazoan is often up to around twice this figure (e.g. *Drosophila melanogaster* has around 15,500, while humans contain around 20–25,000), our annotations are nonetheless a large proportion of the expected gene count of these sponges. This level of annotatable data is not unusual for de novo transcriptome assemblies, which will contain novel genes, fragments of complete genes, non coding RNA and UTR sequences alongside coding sequences. It should be noted that in even the most well-annotated genomes, not all genes can be placed in GO categories.

We also performed KEGG annotation on our de novo transcriptome using the KAAS-KEGG automatic annotation server, with its default BLAST settings, to understand the representation of key pathways in our transcriptomic datasets. The results of these annotations are provided in full as File S3, but in general recovery was excellent with 37,313 genes annotated to existing KEGG terms. While sponges do not possess the full canonical complements of other metazoan phyla, as these were not present in the common ancestors of sponges and the broader Metazoa, we generally recover the expected gene complements of the Porifera in our reference transcriptomic assembly, when mapped to the *A. queenslandica* KEGG dataset. Together with BUSCO results and the raw number of annotated genes, this gives us confidence in the depth of our transcriptomic resources.

## Differential expression results

We utilised our samples to perform a differential expression analysis, aimed at discerning the specific genes up- and down-regulated by exposure to acute increased temperatures. It should be noted that due to experimental necessity, these are few in replicate number compared to many experiments, with $n = 2$ at best. This limits the strength of conclusions that can be drawn particularly between control and treatment conditions, although treatment vs treatment comparisons are more robust. The sample correlation matrix and heatmap of relative expression for differentially expressed genes alone can be seen in Fig. 2. The large degree to which the 3 and 5 °C treatments resemble one another can be seen in the matrix in Fig. 2A, where the four replicates show little differentiation from one another (black regions of matrix). Within-sample variation is observed (the long branches for each terminal node on the dendrogram) but little between-treatment variation is evident (see the small distances between nodes leading to the individual

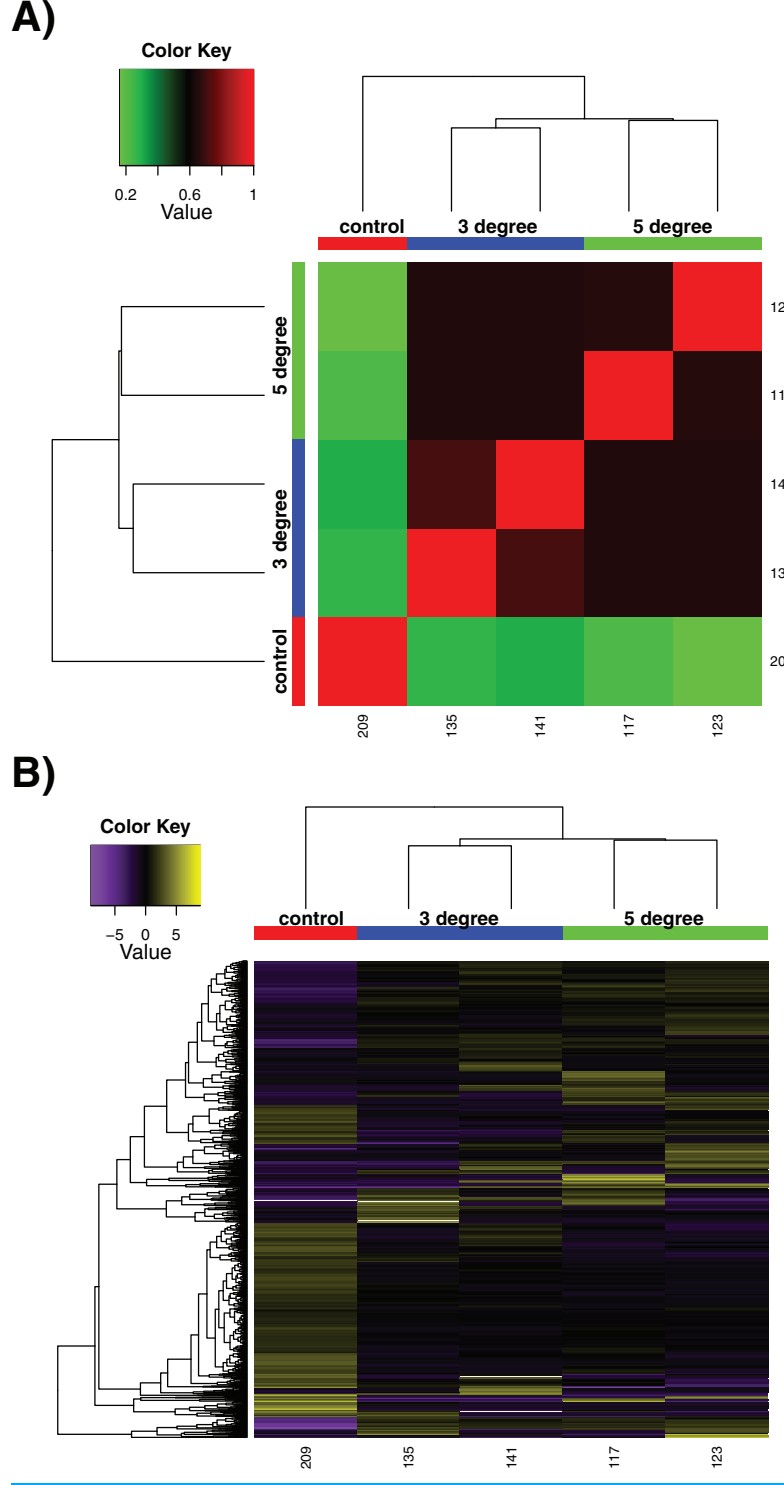

**Figure 2 Differential expression results.** Differential expression analysis results performed by RSEM within the Trinity framework, with 'as gene' results shown. (A) Sample correlation matrix for the five samples used in the final cross-comparison. (B) Relative expression of each differentially expressed contig across all the samples. Note that contigs occurring uniquely in these samples are not included, as detailed in Methods. For both (A) and (B), note the areas of the matrix shown in black hues, indicating few measurable differences between-samples.

replicates). In Fig. 2B, while some discreet inter-sample variation is observed, note the large downregulated quadrant of the matrix at bottom right, relative to high relative expression levels in the control sample at far left. Adding additional replicates would increase confidence in these results.

In general, we found a large number of genes whose expression was perturbed by heat treatment at both 3 and 5 °C (although please note this is in comparison to a single control sample). Between our control and 3 °C treatments, the transcriptional landscape changed considerably. Stringent criteria were applied for significance given the single control: cut off for FDR of 0.0001, a min abs(log2($a$/$b$)) change of 2, but these results should be taken as indicative rather than quantitative. A total of 1,435 contigs were noted to change in expression (767 upregulated, 668 downregulated at the 3 °C treatment compared to the control sample—again note this is a single control). This was mirrored in our control vs 5 °C comparison, where 1,230 contigs changed significantly in expression (623 upregulated, 607 downregulated significantly at 5 °C compared to the single control sample). The complete list of differentially expressed genes and further details are given in File S4. There was a large amount of overlap between these sets. In total, 322 genes were downregulated commonly in both the 3 and 5 °C samples when compared to the control. A total of 383 genes were upregulated commonly in both the 3 and 5 °C samples compared to the control. These genes may be particularly interesting for further analysis, given their conserved role in thermal response, and we have provided these in File S5. Contrastingly, genes that were not shared could provide specific information about responses at specific temperature points, and are also made available for further study. We re-iterate, however, that the single control sample means that these results are not as well-supported as our 3 vs 5 °C analysis.

The 20 most up and down regulated genes (and thus the ones with the highest statistical support, although vs a single control) in our control vs 3 °C and control vs 5 °C samples are of clear interest in understanding thermal stress response, and can be seen in Table 3. These contigs include those with similarity to heat shock proteins (HSP) (*heat shock 70 B2-like*) and ribosomal proteins (*40S ribosomal S13, 40S ribosomal S25*), along with a large number of sequences with no clear homologues in the RefSeq database, and some of less clear utility (e.g. the *Plant Cadmium Resistance 3-like* family of eukaryote-conserved genes, although cadmium and these genes are noted to influence the HSP pathway (*Hofmann et al., 2000*)).

However, very few genes (five upregulated in 5 °C sample vs 3 °C, 11 downregulated) changed significantly in expression between the 3 and 5 °C treatment. This is a more reliable measurement than our control vs treatment comparisons, given the replicates for both of these samples. We speculate that this may be due to the transcriptional machinery of this sponge already operating at maximum capacity to adjust to thermal stress under the 3 °C treatment, with little extra ability to cope with changes brought by further increases in temperature, although longer exposure might also be needed to bring about more discrete transcriptional changes. Despite their small number, these genes are of a variety of annotations, including *allograft inflammatory factor 1, ubiquitin*

**Table 3 The 20 most up/downregulated contigs in each sample cross-comparison.**

| Control vs 3 °C, downregulated | Annotation | Control vs 5 °C, downregulated | Annotation | 3 vs 5 °C, downregulated | Annotation |
|---|---|---|---|---|---|
| TRINITY_DN20177_c1_g1 | NA | TRINITY_DN24223_c2_g2 | PLANT CADMIUM RESISTANCE 3-like | TRINITY_DN23764_c0_g2 | Polyadenylate-binding -interacting 1-like |
| TRINITY_DN33508_c0_g1 | NA | TRINITY_DN20177_c1_g1 | NA | TRINITY_DN24223_c2_g2 | PLANT CADMIUM RESISTANCE 3-like |
| TRINITY_DN45390_c0_g1 | NA | TRINITY_DN8570_c0_g1 | Hypothetical protein | TRINITY_DN12683_c0_g1 | Allograft inflammatory factor 1 |
| TRINITY_DN33508_c0_g4 | NA | TRINITY_DN32712_c4_g3 | NA | TRINITY_DN6163_c0_g2 | Ubiquitin carboxyl-terminal hydrolase isozyme L3-like |
| TRINITY_DN35469_c0_g1 | Soluble calcium-activated nucleotidase 1 isoform X1 | TRINITY_DN24027_c0_g1 | NA | TRINITY_DN23190_c0_g1 | NA |
| TRINITY_DN35123_c0_g3 | NA | TRINITY_DN40223_c0_g1 | NA | TRINITY_DN24468_c0_g2 | NA |
| TRINITY_DN19331_c1_g1 | Hypothetical protein | TRINITY_DN8786_c0_g1 | Oxysterol-binding 1-like | TRINITY_DN30615_c2_g9 | NA |
| TRINITY_DN24815_c0_g1 | NA | TRINITY_DN35469_c0_g1 | Soluble calcium-activated nucleotidase 1 isoform X1 | TRINITY_DN8356_c0_g2 | Ubiquitin-conjugating enzyme E2 K-like |
| TRINITY_DN8271_c0_g1 | NA | TRINITY_DN45390_c0_g1 | NA | TRINITY_DN26078_c0_g2 | COMM domain-containing 8-like |
| TRINITY_DN31187_c0_g1 | NA | TRINITY_DN33246_c0_g3 | NA | TRINITY_DN58418_c0_g1 | Nucleoporin GLE1-like |
| TRINITY_DN49200_c0_g1 | NA | TRINITY_DN33508_c0_g4 | NA | TRINITY_DN35203_c0_g2 | NA |
| TRINITY_DN33462_c0_g1 | NA | TRINITY_DN11747_c0_g1 | Hypothetical protein, partial | | |
| TRINITY_DN4804_c0_g2 | Hypothetical protein TRIADDRAFT_59764 | TRINITY_DN24815_c0_g1 | NA | | |
| TRINITY_DN23863_c0_g1 | NA | TRINITY_DN24481_c0_g1 | NA | | |
| TRINITY_DN33246_c0_g3 | NA | TRINITY_DN30570_c0_g1 | Replicase poly | | |
| TRINITY_DN30570_c0_g1 | NA | TRINITY_DN35123_c0_g3 | NA | | |
| TRINITY_DN26078_c0_g1 | COMM domain-containing 8-like | TRINITY_DN59119_c0_g1 | NA | | |
| TRINITY_DN12307_c0_g1 | PREDICTED: uncharacterised protein | TRINITY_DN13825_c0_g1 | Endoplasmic reticulum-Golgi intermediate compartment 3-like | | |
| TRINITY_DN36675_c0_g1 | NA | TRINITY_DN26662_c0_g1 | Hypothetical protein crov191 | | |

| Control vs 3 °C, upregulated | Annotation | Control vs 5 °C, upregulated | Annotation | 3 vs 5 °C, upregulated | Annotation |
|---|---|---|---|---|---|
| TRINITY_DN24284_c0_g3 | Hypothetical protein BRAFLDRAFT_114823 | TRINITY_DN24284_c0_g3 | Hypothetical protein BRAFLDRAFT_114823 | TRINITY_DN33462_c0_g1 | NA |
| TRINITY_DN44573_c0_g1 | Hypothetical protein | TRINITY_DN24223_c2_g1 | PLANT CADMIUM RESISTANCE 3-like | TRINITY_DN53922_c0_g3 | Inactive peptidyl-prolyl cis-trans isomerase FKBP6-like |
| TRINITY_DN18944_c0_g2 | 40S ribosomal S13 | TRINITY_DN44573_c0_g1 | Hypothetical protein | TRINITY_DN23764_c0_g1 | Polyadenylate-binding -interacting 1-like |
| TRINITY_DN33508_c0_g2 | NA | TRINITY_DN22854_c0_g2 | PREDICTED: tetraspanin-7-like | TRINITY_DN58083_c0_g1 | NA |
| TRINITY_DN24223_c2_g1 | PLANT CADMIUM RESISTANCE 3-like | TRINITY_DN8786_c0_g2 | Oxysterol-binding 1-like | TRINITY_DN23863_c0_g1 | NA |
| TRINITY_DN34845_c4_g12 | NA | TRINITY_DN29050_c0_g1 | Hypothetical protein | | |
| TRINITY_DN34845_c4_g3 | NA | TRINITY_DN18944_c0_g2 | 40S ribosomal S13 | | |
| TRINITY_DN22854_c0_g2 | PREDICTED: tetraspanin-7-like | TRINITY_DN53805_c0_g1 | PREDICTED: uncharacterised protein | | |
| TRINITY_DN34845_c4_g9 | NA | TRINITY_DN30712_c0_g1 | NA | | |
| TRINITY_DN22240_c0_g4 | NA | TRINITY_DN58083_c0_g1 | NA | | |
| TRINITY_DN34845_c4_g10 | NA | TRINITY_DN26231_c0_g4 | Dolichyl-diphosphooligosaccharide–glycosyltransferase 48 kDa subunit-like | | |
| TRINITY_DN44814_c0_g1 | NA | TRINITY_DN27017_c0_g1 | NA | | |
| TRINITY_DN26511_c0_g1 | Hypothetical protein | TRINITY_DN34845_c4_g10 | NA | | |
| TRINITY_DN34845_c4_g4 | NA | TRINITY_DN24395_c0_g1 | Heat shock 70 B2-like | | |
| TRINITY_DN1002_c0_g1 | Predicted protein | TRINITY_DN34845_c4_g4 | NA | | |
| TRINITY_DN23682_c0_g2 | NA | TRINITY_DN34845_c4_g12 | NA | | |
| TRINITY_DN26231_c0_g4 | Dolichyl-diphosphooligosaccharide–glycosyltransferase 48 kDa subunit-like | TRINITY_DN34845_c4_g3 | NA | | |
| TRINITY_DN2258_c0_g1 | 40S ribosomal S25 | TRINITY_DN22240_c0_g4 | NA | | |
| TRINITY_DN30873_c1_g2 | NA | TRINITY_DN26511_c0_g1 | NA | | |
| TRINITY_DN20209_c0_g1 | NA | TRINITY_DN34845_c4_g9 | NA | | |

*carboxyl-terminal hydrolase isozyme L3-like* and a *FKBP6-like* sequence, and could still be helping adjust to thermal stress at specific temperature ranges.

## GO over-representation analysis

To gain a broad understanding of the nature of changes between our samples, we tested differentially expressed genes at given temperatures as our test sets, relative to our reference transcriptomes, and checked for over-representation of any gene categories using Fisher's Exact Test, correcting for false discovery rate and with a cutoff *p*-value of 0.05, as integrated in Blast2GO. Our findings are based on the small number of replicates mentioned previously, but with a high stringency criteria for assigning significance. When we compared our control to our 3 and 5 °C samples a variety of statistically significant differences in contig expression were observed (Figs. 3A and 3B). Please note that as our control sample had only a single replicate, these findings are not as well supported as our 3 vs 5 °C comparison. In the comparison of the 3 vs 5 °C treatments, which do have replicates, there was no difference in significant GO category expression between our 3 and 5 °C samples, and we infer that this is due to only minor changes between 3 and 5 °C. These contigs change little in expression, and from their values at 3 °C do not (or can not) alter their levels any further to adjust to the additional stress at 5 °C.

In our single control vs duplicate 3 °C samples (Fig. 3A), GO category representation changed significantly in a number of categories. Among the GO categories downregulated at 3 °C, 'immune response', 'signal transduction' 'serine type endopeptidase inhibitor activity' and 'pantothenate biosynthetic pathway' were significantly more prevalent there than in our overall set. These GO terms indicate that immune responses and normal bodily maintenance may be perturbed by thermal stress. In contrast, GO terms associated with ribosomes and structural molecules are under-represented in this set, and therefore contigs coding for these are not present in the downregulated complement after heat exposure. This is similar to the results observed in *Guzman & Conaco (2016)*, as discussed further below.

In the GO categories upregulated at 3 °C, it again seems that normal metabolism is less important for response to thermal stress than cytoskeletal responses and growth factor activity. Many GO terms related to metabolic activity (e.g. 'phosphate-containing compound metabolic process', 'organic substance biosynthetic process') are under-represented in these genes relative to our overall library, while GO terms linked to microtubules and other cytoskeletal components are significantly over-represented. Taken in concert with our GO term distribution findings from downregulated genes, it seems that structural componentry is significantly upregulated in response to a 3 °C heat shock, and cytoskeletal proteins can be involved in stress response (*Serafini et al., 2011*).

Many of these findings are mirrored in our single control vs duplicate 5 °C samples (Fig. 3B). Four GO categories, 'intracellular non-membrane-bounded organelle', 'cytoplasmic part', 'pantothenate biosynthetic process' and 'signal transduction', had similar changes in abundance in downregulated contigs from our 3 and 5 °C treatments compared to the control, and 'growth factor activity' was significantly over-represented in contigs upregulated in response to increased temperature in both samples. These have

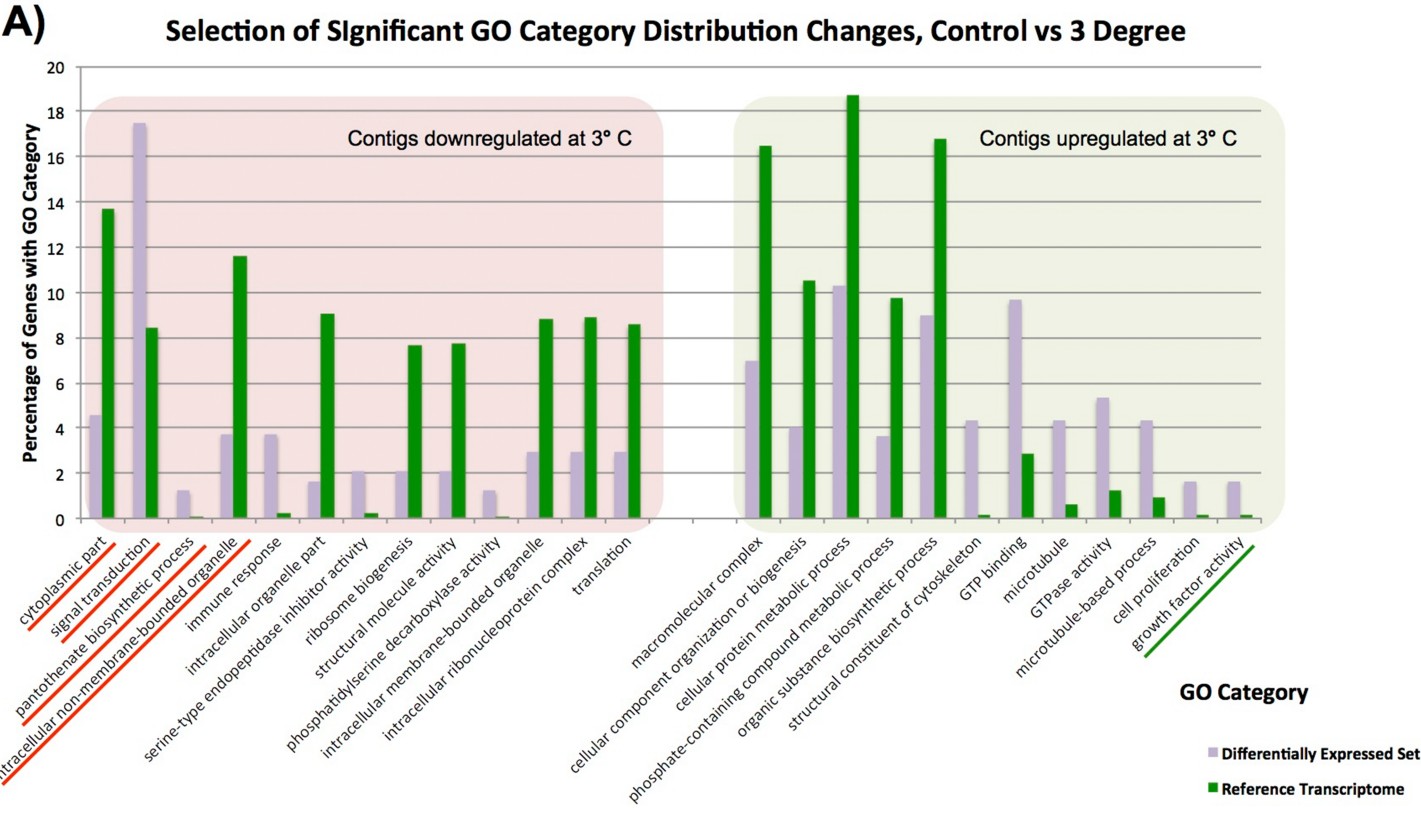

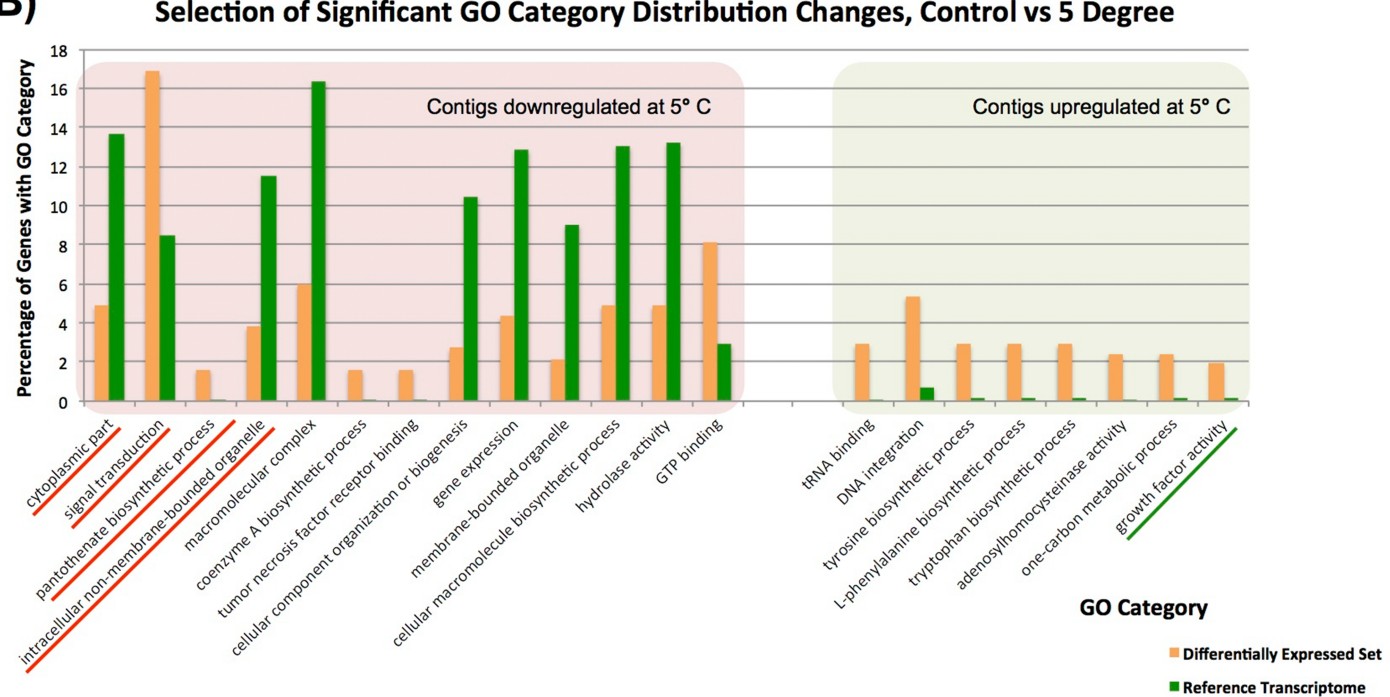

**Figure 3 Significantly differentially represented GO categories.** Significantly differentially represented GO categories from contigs up/down regulated at 3 and 5 °C compared to control sample. (A) Shows categories from our control vs 3 °C comparison, while (B) shows those from the control vs 5 °C comparison. Those categories represented in both comparisons are underlined.

been underlined in the figure. A small number of additional categories were significantly changed in our 5 °C treatment but not observed in our 3 °C treatment, including a number of categories related to the biosynthesis of specific molecular and cellular components. For instance, the category 'cellular macromolecule biosynthetic process', is less prevalent in downregulated genes, (and therefore not itself present in downregulated contigs), while 'tyrosine biosynthetic process' and other categories are over-represented in up-regulated genes, reflecting underlying transcriptional activity in that regard, mirroring our findings in the 3 °C treatment.

Previous work (*Guzman & Conaco, 2016*) observed the upregulation of a number of protective and signalling-related pathways after heat shock. We see a similar pattern in our data, but the exact categories are not seen in *Isodictya* as in their warm-water *Haliclona tubifera*. Protective mechanisms in common with the *Haliclona* results include those linked to antioxidant activity and immune response activation, but do not include the toll-like receptor (TLR) signalling pathway. We also observe similar broad representation of cellular ion homeostasis, messenger-mediated pathways, transporter activity and microtubule-based movement-linked categories, but do not see specifically calcium-mediated signalling categories, unlike those seen in their data.

While most of the observed categories in our data are quite broad and reflect changes in a number of gene families, we can observe some quite specific changes in expression. For example, the 'Tumor necrosis factor receptor binding' category is over-represented in downregulated contigs at 5 °C. This means that TNF receptor binding genes are not as common in the 5 °C treatment, suggesting that this gene family is not utilised as prevalently after heat shock. This finding was born out by our analysis of discrete contig up- and down-regulation, as discussed in detail elsewhere in this manuscript.

## Identification of target genes

Given the large number of genes with no gene ontology assignment in the top 20-most differentially expressed complement, we also adopted a target gene approach to identify the impact of heat exposure on many well-known gene families known from previous work. From the lists of differentially expressed genes, we have selected genes previously identified as playing a role in thermal response for further study, as well as transcription factors with potential roles as cellular mediators of this process, as can be seen in Table 4. Full details of the exact expression levels of all these genes can be found in File S4.

Heat shock proteins were prevalent in these complements. We obtained four contigs encoding a full-length sequence, corresponding to four isoforms of the gene encoding for HSP70 and related proteins. These four sequences were designed as A1, A2, B and HSP-er (contigs: TRINITY_DN27082_c0_g1, TRINITY_DN12545_c0_g2, TRINITY_DN24395_c0_g1 and TRINITY_DN20850_c0_g1_i1, respectively). The alignment of these four putative HSP70 sequences from *Isodictya* sp. alongside those previously described in sponges and other marine invertebrates, as well as related species, can be seen in Fig. 4A (Full sequences, along with alignment, File S6).

The full-length sequence of these proteins showed the presence of three classical signature motifs of the HSP70 family (IDLGTTYS; IIDLGGGTFDVSIL;

**Table 4 Manually identified target genes.**

**Target gene approach**

| Control vs 3 °C, downregulated | Annotation | Control vs 5 °C, downregulated | Annotation | 3 vs 5 °C, downregulated | Annotation |
|---|---|---|---|---|---|
| TRINITY_DN49841_c0_g3_i1 | Tumor necrosis factor ligand superfamily member 15 | TRINITY_DN49841_c0_g3_i1 | Tumor necrosis factor ligand superfamily member 15 | TRINITY_DN23764_c0_g2 | Polyadenylate-binding-interacting 1-like |
| TRINITY_DN49577_c0_g1_i1 | Mesoderm development candidate 1-like | TRINITY_DN49577_c0_g1_i1 | Mesoderm development candidate 1-like | TRINITY_DN24223_c2_g2 | PLANT CADMIUM RESISTANCE 3-like |
| TRINITY_DN34885_c0_g2_i2 | Neurogenic locus notch homolog 1-like | TRINITY_DN34885_c0_g2_i2 | Neurogenic locus notch homolog 1-like | TRINITY_DN12683_c0_g1 | Allograft inflammatory factor 1 |
| TRINITY_DN28743_c0_g1_i1 | E3 ubiquitin-ligase TRIM71-like | TRINITY_DN28743_c0_g1_i1 | E3 ubiquitin-ligase TRIM71-like | TRINITY_DN6163_c0_g2 | Ubiquitin carboxyl-terminal hydrolase isozyme L3-like |
| TRINITY_DN28248_c0_g1_i1 | Ribosome-binding 1-like | TRINITY_DN33363_c1_g1_i1 | G2 M phase-specific E3 ubiquitin-ligase-partial | TRINITY_DN8356_c0_g2 | Ubiquitin-conjugating enzyme E2 K-like |
| TRINITY_DN30424_c0_g1_i1 | Delta and Notch-like epidermal growth factor-related receptor-like | TRINITY_DN28248_c0_g1_i1 | Ribosome-binding 1-like | TRINITY_DN26078_c0_g2 | COMM domain-containing 8-like |
| TRINITY_DN31499_c1_g3_i1 | Calmodulin isoform X1 | TRINITY_DN4806_c0_g1_i1 | Polycomb group RING finger 1-like | TRINITY_DN58418_c0_g1 | Nucleoporin GLE1-like |
| TRINITY_DN29988_c0_g1_i1 | TGF-beta receptor type-1-like | TRINITY_DN30424_c0_g1_i1 | Delta and Notch-like epidermal growth factor-related receptor-like | | |
| TRINITY_DN35391_c8_g6_i2 | TNF receptor-associated factor 5-like | TRINITY_DN31499_c1_g3_i1 | Calmodulin isoform X1 | | |
| TRINITY_DN29288_c0_g1_i1 | G-coupled receptor 161-like | TRINITY_DN35391_c8_g6_i2 | TNF receptor-associated factor 5-like | | |
| TRINITY_DN54189_c0_g2_i1 | Probable G-coupled receptor 157 | TRINITY_DN29288_c0_g1_i1 | G-coupled receptor 161-like | | |
| | | TRINITY_DN54189_c0_g2_i1 | Probable G-coupled receptor 157 | | |

**Target gene approach**

| Control vs 3 °C, upregulated | Annotation | Control vs 5 °C, upregulated | Annotation | 3 vs 5 °C, upregulated | Annotation |
|---|---|---|---|---|---|
| TRINITY_DN12683_c0_g1_i1 | Allograft inflammatory factor 1 | TRINITY_DN25219_c0_g1_i1 | Ubiquitin-ligase E3A-like | TRINITY_DN53922_c0_g3 | Inactive peptidyl-prolyl cis-trans isomerase FKBP6-like |
| TRINITY_DN27456_c0_g1_i1 | Tolloid 1 | TRINITY_DN40009_c0_g1_i1 | Transcription factor Sox-2-like | TRINITY_DN23764_c0_g1 | Polyadenylate-binding-interacting 1-like |
| TRINITY_DN25219_c0_g1_i1 | Ubiquitin-ligase E3A-like | TRINITY_DN24395_c0_g1_i1 | Heat shock 70 B2-like | | |
| TRINITY_DN40009_c0_g1_i1 | Transcription factor Sox-2-like | TRINITY_DN33837_c0_g3_i1 | TNF receptor-associated factor 5-like | | |
| TRINITY_DN24395_c0_g1_i1 | Heat shock 70 B2-like | TRINITY_DN34897_c0_g1_i1 | Tolloid 2 | | |
| TRINITY_DN16789_c0_g1_i1 | Indian hedgehog | TRINITY_DN29751_c0_g1_i1 | Growth differentiation factor 7 | | |
| TRINITY_DN33837_c0_g3_i1 | TNF receptor-associated factor 5-like | TRINITY_DN29790_c0_g3_i1 | Ubiquitin-60S ribosomal L40 | | |
| TRINITY_DN34897_c0_g1_i1 | Tolloid 2 | TRINITY_DN35733_c0_g1_i1 | Probable E3 ubiquitin-ligase partial | | |
| TRINITY_DN29790_c0_g3_i1 | Ubiquitin-60S ribosomal L40 | TRINITY_DN16075_c0_g1_i1 | Bone morphogenetic 7 | | |
| TRINITY_DN28521_c0_g1_i1 | Bone morphogenetic 6-like | TRINITY_DN27508_c0_g1_i1 | Growth differentiation factor 8-like | | |
| TRINITY_DN8356_c0_g2_i1 | Ubiquitin-conjugating enzyme E2 K-like | TRINITY_DN27456_c0_g1_i1 | Tolloid 1 | | |
| TRINITY_DN16075_c0_g1_i1 | Bone morphogenetic 7 | TRINITY_DN34419_c1_g1_i1 | Pinhead precursor | | |
| TRINITY_DN27508_c0_g1_i1 | Growth differentiation factor 8-like | TRINITY_DN30012_c0_g1_i1 | Oxidative stress-induced growth inhibitor 2-like | | |
| TRINITY_DN31167_c0_g1_i2 | Segment polarity dishevelled homolog | TRINITY_DN32735_c0_g1_i2 | G-coupled receptor 161-like | | |
| TRINITY_DN46355_c0_g1_i1 | Hedgehog precursor | | | | |
| TRINITY_DN34419_c1_g1_i1 | Pinhead precursor | | | | |
| TRINITY_DN19433_c0_g1_i1 | MULTISPECIES: cold-shock | | | | |
| TRINITY_DN32811_c0_g1_i1 | Glutaredoxin | | | | |
| TRINITY_DN32811_c0_g1_i3 | Thioredoxin | | | | |
| TRINITY_DN33675_c0_g2_i1 | Alkyl hydroperoxide reductase | | | | |

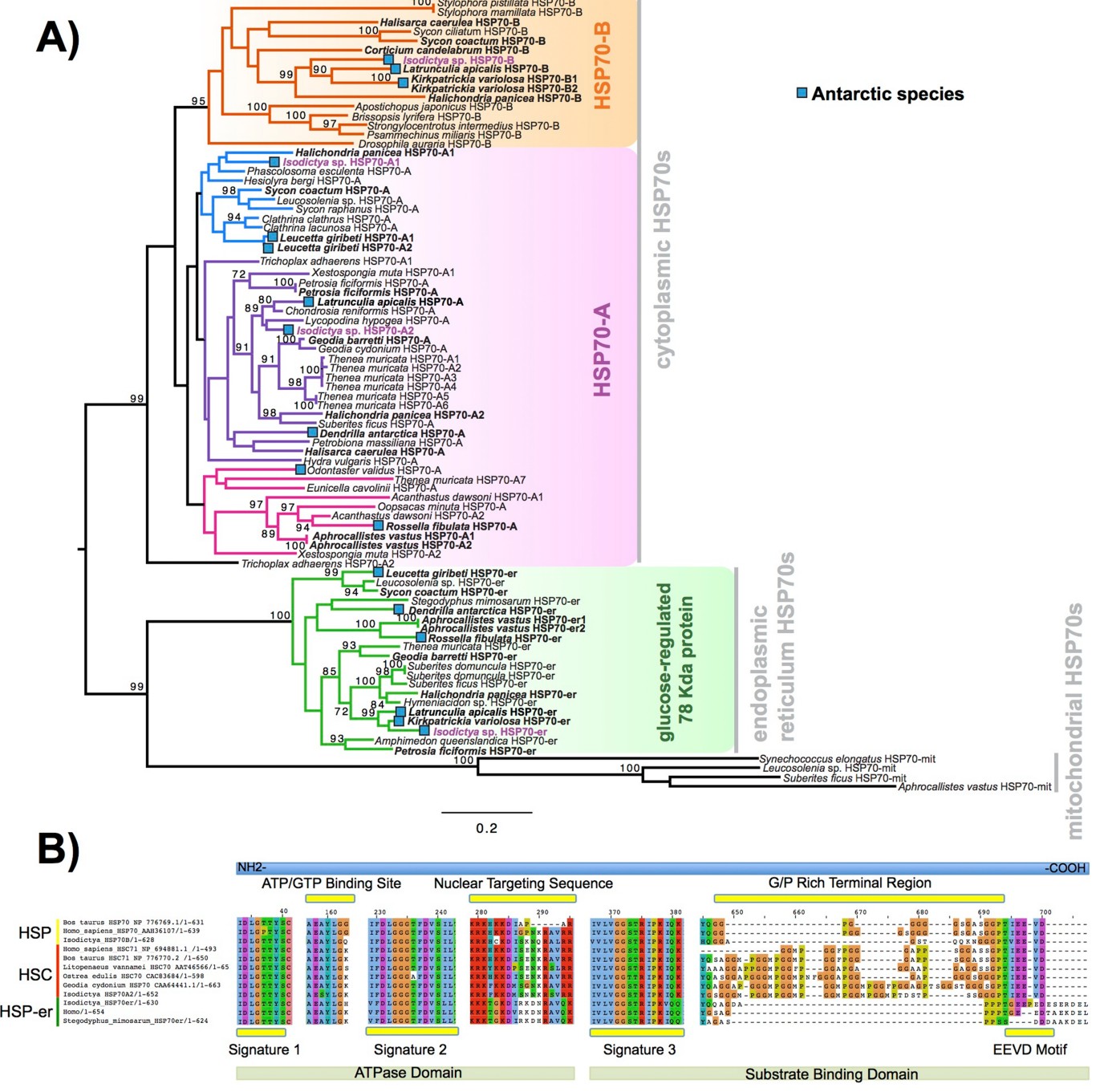

**Figure 4 HSP phylogeny and alignment.** (A) Maximum likelihood-derived phylogeny (RAxML, LG+I+G) of HSP70 sequences of known homology, together with novel *Isodictya* sequences (Names in purple). HSP70 sequences of Antarctic-resident species are bolded. Boxed as indicated on the figure are the HSP70-B, HSP70-A (HSC) and glucose-regulated 78 kDa HSP70 clades. Numbers at base of nodes indicate bootstrap support (as Percentage of 1,000 replicates). (B) Multiple alignment of different HSP70 isoforms found in *Isodictya* sp. with (inducible) HSP70-B, sequence at top, HSP70-A (non-inducible heat-shock cognate) central, and below, HSP-mt sequences. These are shown alongside sequences from other species, showing the differences in domain content between isoforms. All HSPs have two main domains: the ATPase domain involved in ATP hydrolysis, and the substrate binding domain, which binds extended polypeptides, as shown at the bottom of the figure. Yellow boxes represent signature domains within HSP70 proteins. The G/P rich terminal region is prominent in HSPC (HSP-B). The EEVD sequence motif allows the cytoplasmic localisation of HSP and HSC isoforms. Accession numbers as provided in Figure, all sequences are provided, along with alignments, in File S6.

IVLVGGSTRIPKI/VQK) and the ATP/GTP binding site (Fig. 4B). These four forms of HSP70 in *Isodictya* possess relatively low sequence similarity (between 41.4% and 68.1% amino acid identity), and individual isoforms are more closely related to some previously described forms, which may indicate that HSP70A and B are ancestrally shared paralogues within the Porifera (Fig. 4A). The HSP70A1 and two genes are most closely related to the non-inducible or HSP70 cognate (HSC) described for others invertebrates and vertebrates (Fig. 4; File S7). Their C-terminal region contains five repeats of GGMP motif. The tetrapeptide motif GGMP repeats are thought to be involved in co-chaperone-binding activities of constitutive forms (*Cascella et al., 2015*). HSP-A sequences have a terminal motif (EEVD) indicating their cytoplasmatic localisation (*Shonhai, Boshoff & Blatch, 2007*). We do not observe a clear mitochondrial-located HSP70 homologue, but HSP-er shows clear homology to known endoplasmic reticulum isoforms, and clear sequence similarity to those from both vertebrate and invertebrate species (Fig. 4B).

In contrast, the isoform HSP70B showed high sequence similarity to the HSP70 sequence of *Xestospongia testudinaria* (75.4%) and this sequence does not possess tetrapeptide repeat sequences at the C-terminal (Fig. 4B). HSP-B is therefore likely the 'inducible' HSP gene in this species. It was up-regulated during thermal stress in this experiment, listed among the most strongly-changed in expression (Table 3), and thus shares characteristics of inducible isoforms previously described in other marine invertebrates such as oysters and krill (see *Cascella et al., 2015*; *Piano et al., 2005* for more details).

In other species, the transcription of HSPs is upregulated in response to a variety of stresses (*Craig & Gross, 1991*), preventing the mis-folding and aggregation of proteins, but is notably absent in some species of Antarctic fish where HSPs are constitutively turned on in some cases (*Shonhai, Boshoff & Blatch, 2007*). HSPs however play a clear role in temperature response in Antarctic invertebrates (*Clark et al., 2017*). Given the upregulation of isoform B here, it is clear that this role is conserved in sponges, and this collaborates findings previously in this phylum (*Pita et al., 2013*; *Krasko et al., 1997*).

A number of contigs with sequence similarity to the *Tumor necrosis factor* gene family were observed in our dataset, in both our upregulated and downregulated complements (n.b. different contigs were present in the two complements). This finding dovetails with the observation of this in the GO term over-representation analysis for contigs downregulated in our 5 °C treatment. This cytokine family is responsible for regulating cell death through ubiquitins, but also has other functions, including the regulation of cell survival, proliferation and differentiation (*Vilcek & Lee, 1991*; *Wang & Lin, 2008*) and the downregulation of some members of this class and upregulation of others possibly reflects changing cellular processes within our sponge samples of interest to those studying the molecular processes underlying thermal stress responses. A single gene linked to this has previously been observed to be upregulated after heat shock in sponges (*TNFAIP3* (*Guzman & Conaco, 2016*)) and the additional sequences observed here suggests the possible role of these genes in linking with ubiquitins and mediating cell death and protein degradation in Porifera after heat exposure.

The ubiquitin related genes themselves, which play essential roles in protein degradation (*Glickman & Ciechanover, 2002*; *Mukhopadhyay & Riezman, 2007*), were well recovered in our transcriptome. Figure 5 shows our annotation of the proteolytic pathway mediated by these genes, as annotated by bi-directional best blast hit. Of genes that are missing, few (12) may be truly absent from our transcriptomic resource—others are also absent from *A. queenslandica* and may be absent ancestrally. Three components of this pathway, TRINITY_DN31115_c0_g1 (*UBE2M; UBC12; ubiquitin-conjugating enzyme E2 M*), TRINITY_DN8356_c0_g2 (*UBC1; ubiquitin-conjugating enzyme*, aka *HIP2; huntingtin interacting protein 2*) and TRINITY_DN25219_c0_g1 (*UBE3A; E6AP; ubiquitin-protein ligase E3A*) were significantly upregulated in our dataset in response to heat exposure. These genes, representing E2 and HECT E3 type ubiquitin conjugating genes could play specific roles in the degeneration of misfolded and degenerated peptides after heat exposure, correlates well with previous findings in other sponge species such as *H. tubifera* (*Guzman & Conaco, 2016*) and the specific genes are worth further consideration as potential mediators of this intracellular protective response.

We found a number of notable transcription factor and cell signalling pathways in our up- and down-regulated gene lists, including genes in the Notch, Wnt, TNF, Sox and TGF-β pathways. These are sometimes recovered in heat stress studies, with the exact families used varying according to study species as detailed below. These genes do not need to be the most up-/down-regulated genes to have a noted effect on transcription, and they represent intriguing targets as possible controllers of wider processes of molecular adaptation.

Other examples of key cell signalling molecules identified in our study include members of the Sox and Wnt gene families, notably *Sox2* and *dishevelled*. These are possibly key in the control of growth response to heat stress—*Sox2* in particular is known for its role in the control of pluripotency in stem cell lineages (*Rizzino, 2009*). Growth will both be promoted by higher temperatures, but also necessary for repairing tissue damage caused by exposure to deleterious conditions. However, by far the most commonly observed family of cell signalling molecules are representatives of the BMP/TGF-β pathway. A variety of contigs with blast similarity to the TGF-β family of signalling molecules were observed in our differentially expressed complements, including a variety of contigs annotated as bone morphogenic proteins and growth differentiation factors, as well as *one eyed pinhead* and the *Tolloid metalloprotease*, which act to modulate TGF-β signalling. These molecules are well known as modulators of cell fates, but any role in controlling response to thermal stress has not before been noted in sponges. It is possible that these molecules are acting downstream of HSP70 (as noted by *Yao et al. (2009)* and *Lee et al. (2016)*) or as part of the control of molecular response to physical damage caused by high temperatures (*Camus et al., 2005*). However, the phylogenetic distance between the organisms where this has been studied and sponges is vast, and more targeted analysis is necessary to discern the true roles of these crucial molecules in this Phylum.

We noted the presence of the Hedgehog family of genes in our dataset, with both *hedgehog precursor* (TRINITY_DN46355_c0_g1_i1) and *indian hedgehog* (TRINITY_DN16789_c0_g1_i1) automatically annotated—the latter will be a naming

# Ubiquitin Mediated Proteolysis

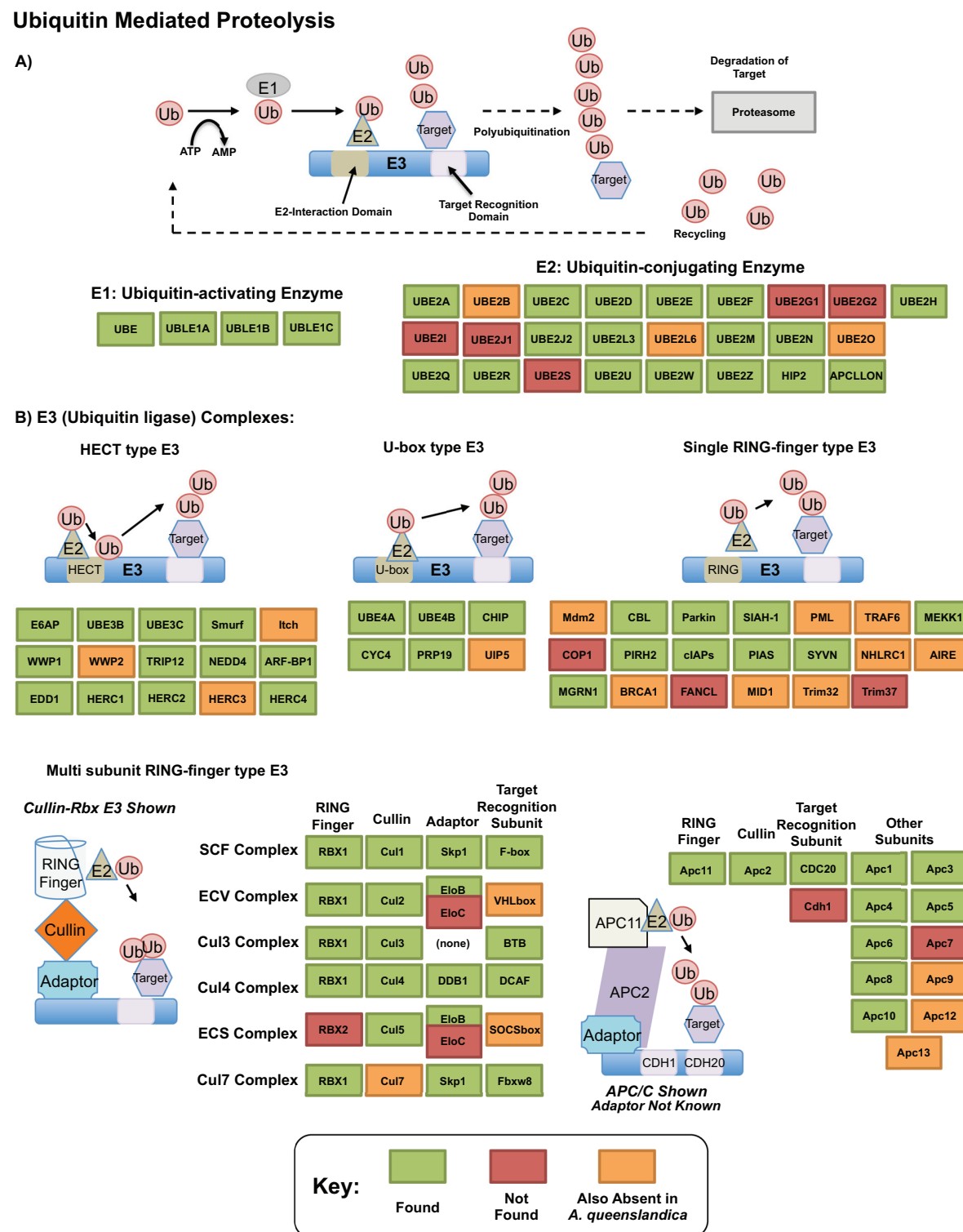

**Figure 5 Ubiquitin pathway recovery.** KEGG style map showing recovery of *Ubiquitin* mediated proteolysis genes in our resource as assessed by bi-directional best blast hit, alongside that of the sequenced genome of *Amphimedon queenslandica*. Genes noted as present in this resource indicated in green, with those absent in our transcriptome noted in orange if also missing from *A. queenslandica* (and therefore a possible poriferan absence), or red if absent from our resource only (and therefore a likely true absence). (A) E1 and E2 mediated proteolysis. (B) E3 mediated proteolysis pathways.

artefact from annotation, as *indian hedgehog* is a chordate novelty generated by duplication into paralogues. Both contigs therefore represent variants of the *hedgehog* gene itself. These could perform key roles in the regulation of response to thermal stress, as they are instrumental in the fine-tuning of cell differentiation responses (*Ingham & McMahon, 2001*). We also note the presence of members of the Notch signalling cascade in the differentially expressed complements following heat stress (e.g. TRINITY_DN30424_c0_g1_i1 *delta and Notch-like epidermal growth factor-related receptor-like*), with Notch-related contigs downregulated after exposure to both 3 and 5 °C treatments. This possibly reflects changes in how cell differentiation is occurring. *Hedgehog* is often noted as a promoter of stem cell proliferation, and the initial stages of heat stress response may involve this, in concert with genes such as *Sox2* as noted above.

In the most similar study to this work previously performed, in the warm water sponge *H. tubifera* from Philippines (*Guzman & Conaco, 2016*), multiple *G protein coupled receptors* (GPCRs) were noted as differentially expressed. In that experiment, sponges with a normal growth temperature of 29 °C were exposed to 32 and 34 °C for between 4 and 12 h. GPCR genes are posited to allow these sponges to monitor and respond to their environment effectively. Several families of GPCR were noted as effected by heat exposure in *H. tubifera*, including *glutamate*, *rhodopsin* and *secretin GPCRs*. None of these were observed here, and therefore the light and contraction responses seen in *H. tubifera* on exposure to heat stress do not seem to be conserved in *Isodictya* sp. Instead, two families of orphan GPCR, *157* and *161*, were noted in our sample. The roles of these specific GPCR in sponges, and in general outside the Chordata, are poorly understood, but their marked up-regulation here, and the absence of the plethora of differentially expressed GPCR seen in *H. tubifera*, suggests that these explicit genes may play specific roles in sensing or recovering from heat exposure in this particular species.

Marine polar organisms are interesting models for studies on the noxious effects of reactive oxygen species (ROS) in cold waters. In order to cope with the formation of deleterious ROS, some Antarctic marine invertebrates express enzymes and low-molecular weight 'scavengers' (*Camus et al., 2005*). The accumulation of ROS during heat stress has been shown in previous studies on Antarctic marine species (*Abele et al., 1998*; *Heise et al., 2003*), and are important to these species given the higher levels of dissolved oxygen seen in cooler Antarctic waters. In our study, we have obtained a good sequence coverage for several enzymes implicated in antioxidant defense (File S8). Two kinds of *superoxide dismutase* (SOD), each with different isoformal variants, were identified in *Isodictya* sp. which are differentiated by the type of metal associated with its active site of the enzyme: two contigs for *manganese SOD* (TRINITY_DN32241_c0_g1_i1 and TRINITY_DN3280_c0_g2_i1) and four contigs for a *copper/zinc SOD* (TRINITY_DN22531_c0_g1_i1; TRINITY_DN17486_c0_g1_i1; TRINITY_DN_65_c0_g1_i1; TRINITY_DN35772_c0_g1_i1). The SOD enzyme catalyse the first reaction in the dismutation of superoxide anions to water and hydrogen peroxide, before the catalase enzyme transforms the hydrogen peroxide into water. This enzyme is represented by three contigs that share high sequence identities with marine invertebrates such as molluscs (TRINITY_DN_6504; 28381; 29099). Another transcript well represented in our dataset was a *glutathione S-transferase* (GST) which plays a key role in the detoxification of

ROS and the regulation of redox balance (*Kim et al., 2009*). In *Isodictya* sp. seven contigs were identified with high sequence similarity to *GSTs*, a complex family of genes with multiple intracellular functions. Previous studies have highlighted the upregulation of oxidative response and antioxidant genes in response to thermal challenge in sponges, as also observed in corals (*DeSalvo et al., 2008*). In our data, we observe the upregulation of several contigs with strong similarity to genes with antioxidant activity, *glutaredoxin* (TRINITY_DN32811_c0_g1_i1) and *thioredoxin* (TRINITY_DN32811_c0_g1_i3) (*Guzman & Conaco, 2016*). We also see a contig upregulated with similarity to *oxidative stress-induced growth inhibitor 2-like* (TRINITY_DN30012_c0_g1_i1), which could play a role in downstream organisation after the detection of an oxidative stress state. It seems possible that these sponges are under severe oxidative stress, and the expression of these genes reflects this, although the experiment was stopped before any outward signs of damage (e.g. necrosis) were observed.

The presence of *allograft inflammatory factor* (TRINITY DN_12683_c0_g1_i1) in our datasets was noted as these have been previously studied in Antarctic species such as the sea urchin *Sterechinus neumayeri* (*Ovando et al., 2012*). The full *Isodictya* AIF-1 sequence was recovered, and has a high sequence similarity with that known from the sponge *Suberites domuncula*, and has two conserved calcium binding motifs known as EF hands (Fig. 6). AIF-1 sequences from several invertebrate and vertebrate species were aligned and showed relatively high levels of conservation of the protein with both groups. Phylogenetic analyses generated a congruent tree positioning the AIF from sponges in a cluster with corals, distinct from vertebrate sequences. In general, *allograft inflammatory factor* is known for its role in recovery from injury, rather than any specific role in protecting against heat shock (*Utans et al., 1995*). It has been studied in sponges previously (*Kruse et al., 1999*), where it has been shown to act to activate immunocyte-like activity, as well as in protection immediately after trauma. In the Japanese oyster *Crassostrea gigas* AIF-1 stimulates hemocyte immune activation by enhancing phagocytosis and expression of inflammatory cytokines (*Zhang et al., 2013*). Its role in the thermal stress response might therefore be in protection against infection, rather than directly in tissue repair, but both these processes are undoubtedly necessary under prolonged periods of acute thermal stress.

That Antarctic species of sponge utilise many of the same pathways to respond to thermal stress as warmer water species is a useful if unsurprising finding. Their efficacy at moderating response to different levels of stress does however seem to be diminished, with many of these genes expressed at both 3 and 5 °C, with no significant difference in their expression levels between these temperatures, as noted elsewhere in this work.

## Poriferan responses to temperature stress

Our results, when considered alongside previous findings in sponges from other latitudes, corroborate previous conclusions regarding their transcriptomic response to acute thermal stress, while suggesting specifically that cold-adapted sponges may have a limited range of tolerance to increased temperatures. The changes in transcription which occurred in our samples after acute short-term thermal exposure suggested that a poriferan heat-response

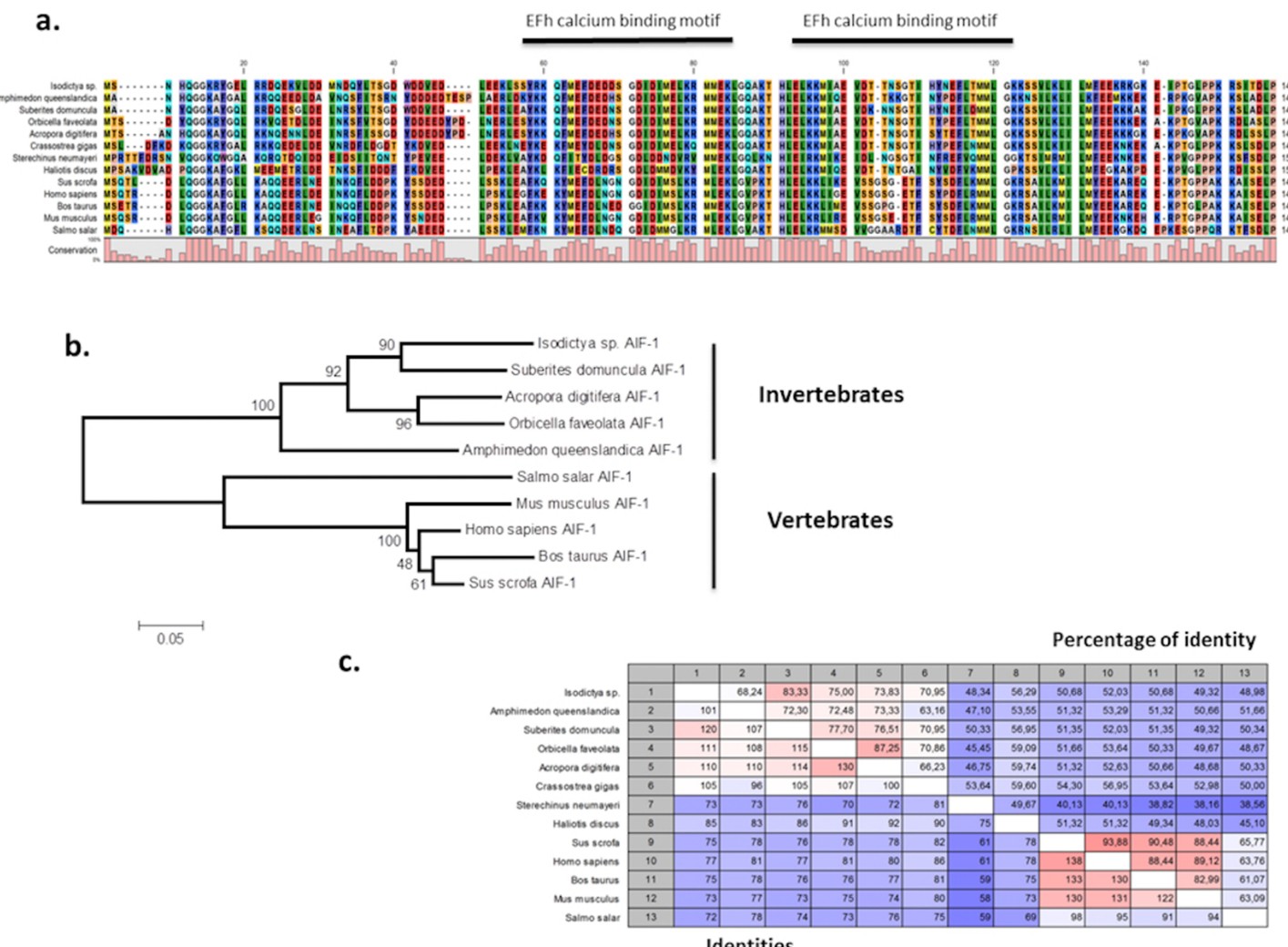

**Figure 6 Alignment and phylogenetic analysis of the AIF sequences found in our transcriptome.** (A) The conserved two EF-hand motifs are indicated in the alignment of several AIF-1 from other invertebrate and vertebrate sequences; (B) Phylogenetic tree of the AIF-1 family using the N-J method. Bootstrap values represent the frequency of appearance (expressed as a percentage) of each clade of 1,000 bootstrap replicates; (C) Percentage amino acid identities between AIF-1 genes from invertebrates and vertebrates. The GenBank accession numbers for the sequences are as follows: *Suberites domuncula* CAC38780; *Acropora digitifera* XP015755194; *Orbicella faveolata* XP020615825; *Amphimedon queenslandica* XP003387413; *Salmo salar* AC169994; *Mus musculus*; *Homo sapiens* P55008; *Bos taurus* NP001071547; *Sus crofa* P81076; *Crassostrea gigas* NP001292275; *Sterechinus neumayeri* ACO40483; *Haliotis discus* ABH10674.

is activated by increased temperatures, and is present in this species. The complements of all four replicates in both 3 vs 5 °C conditions were very similar in both overall pattern and individual composition, with differentially regulated contigs often perturbed in common at both temperatures. Previous work on sponges from other latitudes (*Guzman & Conaco, 2016*; *Gouy, Guindon & Gascuel, 2009*) showed that HSP70 and ubiquitin-related genes were specifically upregulated in response to thermal stress. Our work echoes this finding, observing the activation of similar genes, such as oxidation-stress related genes. The nature of the thermal stress response therefore is broadly similar to that seen in other sponge species. What may differ in Antarctic species of sponge is the limits of tolerance.

In other sponges, the degree and type of transcriptional response is temperature-dependent, with specific genes activated at different temperatures (*Fan et al., 2013*), where the holobiont changes, or (*Guzman & Conaco, 2016*), where the sponge itself adapts its transcription). In contrast, our findings showed few significant differences in the transcriptional response of sponges at 3 and 5 °C (where we have replicate samples, and can thus be confident of the veracity of these results). We also do not see changes in quite the same genes, with the absence of differentially expressed GPCR and TLR signalling genes in our data a notable difference. While previous research has demonstrated that some Antarctic species are not able to respond to additional increases in seawater temperature, others, such as *Nacella concinna* or *Laternula elliptica* showed a classic thermal stress response with over-expression of heat stress chaperones.

A number of differences are noted in the response of cold water sponges here, when contrasted with those of warm-water species (*Guzman & Conaco, 2016*). Particularly, there does not seem to be the degree of incremental change that can be seen in the transcriptional profile of warm water species on exposure to increasing temperatures, at least in the time frame examined here. Since our experiment exposed sponges to acute stress, our results did not test whether cold-adapted sponges could, in time, adapt to sequentially higher temperatures, and this would be a useful follow-up experiment to those described here. It may be possible that with a period of acclimation, sequential rises in temperature can be met by successive rounds of molecular adaptation. However, at present we cannot speculate as to whether that is possible in this species.

## CONCLUSIONS

Climate change is a pressing issue globally, and will have a range of effects on organisms worldwide, involving both long-term and acute exposure to increased temperatures. While some of these effects are more obvious or better studied than others, it is some of the least-studied species which will bear the brunt of these conditions, and these will have a variety of knock-on effects. Here we have studied one such species, a locally abundant sponge, specialised to life in Antarctic conditions. We have generated a reference transcriptome from a small number of replicate samples exposed to acute thermal stress, and identified the molecular responses that these species use to adjust to these conditions. Further, we have demonstrated the clear stress that even a modest increase in temperature over a short time frame (in this case, 4 h) will place on such a specialist species.

Using this data, we have made several comparisons between a single control sample and replicate 3 and 5 °C heat shock samples. Comparisons between our replicate samples indicate *Isodictya* already exercises a full measure of transcriptomic response to ambient temperatures of 3 °C, and further stress at 5 °C leaves it little further 'wriggle room', at least when considering acute responses such as those tested here. Whether this stress results, in the wild, in the death of these sponges, or whether, given time, they will adapt to increased temperatures, is at present uncertain. However, in the interim the results presented here will allow us to begin to understand the impact of increased temperatures on these still under-investigated, but nonetheless vital, species.

## ACKNOWLEDGEMENTS

The authors thank the members of their laboratories for all their support, comradeship and scholarly input. We are also grateful to the dive team, Leslie Novoa, Juana Levihuan and INACH personnel at Yelcho Station for their help during fieldwork activities in Antarctica. Cristian Lagger provided the underwater photograph of *Isodictya* sp. We thank the editors and reviewers for their aid in assessing this manuscript. This paper contributes to the SCAR Antarctic Thresholds—Ecosystem Resilience and Adaptation (Ant-ERA) programme.

### Funding

César Cárdenas was supported by CONICYT/FONDECYT/INACH/INICIACION/ #11150129. This study was partially funded by the INACH program "Marine Protected Areas". Nathan Kenny and Ana Riesgo and this project received funding from the European Union's Horizon 2020 research and innovation programme under the Marie Sklodowska-Curie grant agreement No. 750937. The funders had no role in study design, data collection and analysis, decision to publish, or preparation of the manuscript.

### Grant Disclosures

The following grant information was disclosed by the authors:
CONICYT/FONDECYT/INACH/INICIACION/#11150129.
European Union's Horizon 2020 research and innovation programme under the Marie Sklodowska-Curie grant agreement No: 750937.

### Competing Interests

The authors declare that they have no competing interests.

### Author Contributions

- Marcelo González-Aravena conceived and designed the experiments, performed the experiments, contributed reagents/materials/analysis tools, prepared figures and/or tables, authored or reviewed drafts of the paper, approved the final draft.
- Nathan J. Kenny performed the experiments, analysed the data, prepared figures and/or tables, authored or reviewed drafts of the paper, approved the final draft.
- Magdalena Osorio performed the experiments, authored or reviewed drafts of the paper, approved the final draft.
- Alejandro Font performed the experiments, authored or reviewed drafts of the paper, approved the final draft.
- Ana Riesgo analysed the data, contributed reagents/materials/analysis tools, prepared figures and/or tables, authored or reviewed drafts of the paper, approved the final draft.
- César A. Cárdenas conceived and designed the experiments, performed the experiments, contributed reagents/materials/analysis tools, prepared figures and/or tables, authored or reviewed drafts of the paper, approved the final draft.

## Field Study Permissions

The following information was supplied relating to field study approvals (i.e., approving body and any reference numbers):

The study was conducted under the permit 806/2015 granted by the Chilean Antarctic Institute (INACH).

## Data Availability

The raw data is available at Figshare: Kenny, Nathan (2019): Supplementary Files, *Isodictya* sp transcriptomics. figshare. Dataset. DOI 10.6084/m9.figshare.7048727.

It is also available at NCBI SRA: PRJNA415418.

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
