# Peer review of "Warm temperatures, cool sponges: the effect of increased temperatures on the Antarctic sponge Isodictya sp"

_PeerJ, doi:10.7717/peerj.8088_

## Round 0.1 · original submission · Major Revisions

Please see the concerns of Reviewers 1 and 2 specifically and especially their notes about lack of replication. Please address specifically what the inference is given the lack of replication and why and how the results can be informative despite this critical problem with the design. That argument will be critical for consideration of acceptance.

·

Basic reporting

The language is clear and unambiguous.

Introduction should include more detail about the choice of study organism and the temperature conditions experienced at the collection site.

Figures are relevant and are discussed well in the text.

Experimental design

Methods are detailed and well described.

Validity of the findings

Data are provided as supplemental information. A link to a table showing all available annotations and expression values for contigs in the reference transcriptome would be useful for corroborating results described in the manuscript. The transcriptome described here is a valuable addition to available sponge sequence resources.

Statistical analyses and thresholds used are suitable for low/no replication. However, the lack of a control replicate seriously limits confidence in the validity of the identified differentially expressed genes, as well as the strength of conclusions that can be drawn from the data. Therefore, interpretation of results needs to be carefully considered and limitations must be clearly stated where applicable.

While the analysis provides some interesting insights on the temperature response of this species, improvement of the annotation pipeline may allow more sequences to be assigned and included in the interpretation of temperature-responsive functions.

Additional comments

Line 87: Additional information on Isodictya is needed. In particular, why was it selected to test the effects of warming in the WAP? Also, what is the typical temperature range experienced by the sponge in its natural habitat?

Line 174 and 268: Were transcripts clustered after assembly to minimize redundancies?

Line 291: Very few genes (~23%) were annotated by blastp alignment at an evalue cutoff of 1x10^-6. This has resulted in much of the data being disregarded from the analysis. Have the authors tested annotation at less stringent evalue cutoffs to find a suitable setting that would allow more sequences to be annotated without necessarily compromising specificity?

Line 293: Were the Isodictya sequences compared to other available sponge sequences, particularly other cold-adapted species? Any similarities might provide clues to the repertoire of genes that are critical for survival in this type of environment or genes that may respond to temperature perturbations.

Line 309: Was domain annotation performed for all predicted peptides? What percent of the predicted peptides have recognizable protein domains. Analysis of domains present in differentially expressed transcript sets would also help in inferring temperature-responsive cellular pathways even in the absence of gene ontology annotation. A summary figure or table for all annotation data would be useful to show how much of the assembled transcriptome was translated, how many have recognizable domains, how many have matches against various databases, and how many have annotation assignments.

Line 313: Annotated peptides make up only 23% of the reference transcriptome (or ~41% based on KEGG annotation), which means that majority of the sequence data (~77%) is disregarded from analysis. How does this compare to other de novo assembled transcriptomes?

Line 333-334: The statement that “replicate samples at 3C are almost identical but there was a slight fluctuation in the 5C samples” is vague and is not apparent from the figures. Please rephrase.

Line 335: Clarify if figure 2A show the correlation based on the full transcriptome profile of each sample or from expression of just the differentially expressed genes.

Line 340: The major difference apparent in figure 2B is that some genes are present in the control but less so in the heated samples. It’s also apparent that there is high intercolony variability in terms of transcript expression. Thus, if there were another control replicate, the set of differentially expressed genes would very likely change and this should be emphasized here. Also, please revise the statement that “the initial findings are already compelling.”

Line 363: Could the similarity between the 3C and 5C treatments be due to the short duration of the treatment (4hr) rather than due to the transcriptional machinery operating at max capacity?

Line 416: “their” to “the”

Line 424: What would it mean for downregulation of TNF receptor binding genes at elevated temperature?

Line 429: This section is quite interesting. Perhaps the section header should read ‘target genes’ instead of ‘target signaling cascades’ to more broadly cover the types of genes discussed.

Line 430: Rather than “un-annotatable genes” one might say “genes with no gene ontology assignment.” Note that the number of annotatable genes likely depends on the stringency thresholds applied. This is why it would be useful to look at top refseq blastp hits at varying evalue cutoff.

Line 496-500: Need more detail on how these genes relate to the temperature response and processes of molecular adaptation. Are these usually identified in heat stress studies?

Line 501-512: This paragraph could be moved elsewhere as it seems out of place between paragraphs discussing transcription factors and morphogens.

Line 588-591, 617-619, 627-619, 639-639: It’s an interesting idea that cold-adapted organisms have an “all or nothing” response to warming. However, it would be good to provide alternative explanations for why not much difference is seen between these temperature treatments. Are 3C and 5C within or outside the normal range experienced by the sponge in the WAP? If these are both extreme temperatures for the sponge, perhaps the molecular and cellular effects of 3C or 5C exposure would be similar. As noted previously, the similarity in the response of the sponge to 3C and 5C treatments could also be due to the short duration of the experiment, rather than a diminished ability to deal with high temperature exposure, and should be indicated.

Line 632: Remove “excellent”

Line 634: change “begun to identify” to “identified” and remove “will”

Figure 2: Please check legend. Based on the methods, computation of expression values was done by RSEM but differential expression analysis was conducted using edgeR.

Reviewer 2 ·

Basic reporting

DNA data checks: fulfilled
Field Study permits: fulfilled
Basic reporting:
This manuscript is well written, with a very good standard of English. In places the manuscript is over-written, in terms of excessive explanation or over-justification given in some places, but specific examples will be listed in general comments below. The referencing is generally OK, but I have a few suggestions as to more suitable references in certain places (again noted in general comments below). There are a lot of references (90), which is rather excessive for this type of manuscript and could be cut back a little to, I would suggest 60-70 at most (which is recommended by a lot of journals, such as Global Change Biology). I appreciate this is sometimes difficult, but it does help to critically evaluate whether all the references are needed and I would encourage the authors to cut back a few references. The structure of the manuscript conforms to PeerJ standards, but fails with respect to statistical rigour.

Experimental design

Experimental Design:
The authors, correctly state this this experiment is an acute heat shock experiment. However, they then go onto discuss this with relevance to IPCC predictions and climate change. These are simply not equivalent. With a 4 hour experiment, the animals could still be in a state of resistance and this does not allow extrapolation to long-term chronic responses to climate change. The authors are correct when they state that little is known about responses to change in Antarctic species, therefore there is some novelty in these data, particularly with regard to the presence of a heat-shock response. What they should have done, which is much more appropriate is to compare these data with other acute heat shock experiments in temperate and tropical corals: do you get the same responses? Is something different? However, this does require similar acute experiments to have been performed in other coral species.
On a more serious note, which the authors admit: there is insufficient replication in this experiment with only 1 control and 2 each of the temperature treatments. This somewhat invalidates the statistical analyses, especially as there looks to be quite a few differences between individuals at 3°C and 5°C and they are only comparing against ONE control sample. There is clearly variation between individuals and at least n = 3 should have been used for each treatment in this experiment. This is a serious experimental flaw and significantly reduces the quality of the output and validity of the findings.

Validity of the findings

I appreciate that a lot of work has gone into this article, but the limitations of the experimental design are not reflected in the results and discussion. The latter would work much better if focussed specifically on comparisons of similar experiments performed in temperate and tropical sponges, not climate change.

Additional comments

Specific comments:
L60. Ref 9 rather outdated and both are best replaced by Turner et al (2016) Nature535, 411-415
L61-64: description of IPCC 2001 results not necessary, as replaced by IPCC 2014.
L69: Suggest ref 14 replaced by Somero (2010) JEB 213, 912-920 which is more appropriate as he refers to importance of long-lived species, such as those in the Antarctic.
L82-85: duplication of line starting on line 64.
L86: refs 17,28,29 are not the best as these do not refer to processes over evolutionary time.
L92-97: not needed, as we are now all fully aware of the power of NGS.
L110-112: statement not justified as you cannot use results from a single 4 hour heat shock to predict chronic responses.
L139-141: again erroneous reference to comparability to climate change responses.
L236-242: No need to describe results or samples not included in the study.
L256: reference transcriptome: surely need to refer to Table 2 here?
L267-278: over wordy reads rather like a thesis including justifications. You can just refer to Table 2 which has all the relevant data. Reference to samples having little genetic variability not needed given sample sizes are severely limited.
L279: suggest you add BUSCO output to supplementary tables to avoid having to explain in detail (or add as a manuscript figure).
L285-287: no need to justify results in results section.
L292-299: not really needed. Lack of sequence similarity to other species almost certainly due to lack of closely related sequenced species.
L308-309: you can just refer to the table.
L311-315: again superfluous explanation.
L330-332: acknowledgement of lack of replication: a really critical issue.
L33: mention of “replicate” samples and correlation: there look to be quite big differences between samples in Fig 2B. Not sure that Fig 2A really shows anything, especially as 2B is far more detailed. Fig 2A could easily be deleted.
L337-339: disagree: there is variation between samples.
L340: disagree results are not compelling with lack of replication.
L356-362: No pint including Table 3 as most sequences have no annotation. This does not help with result interpretation.
L363-370: lack of difference between 3°C and 5°C is interesting, but given such little differential expression, little significance can be placed on the annotation of those genes.
L399: cytoskeletal proteins can also be involved in the stress response: see Tomanek et al. (2012) JEB 215, 3905-3916 and Serafini et al (2011) Comp Biochem Physiol D 6, 322-334 as examples.
L384-413: just a description of GO categories which are very vague and do not explain an awful lot and Fig 3 not very informative. Much better to concentrate on specific biological results coming out of data.
L430: disappointing to see emphasis on candidate gene approach as RNA-Seq is supposed to be discovery –led. Needs better justification.
L461: see Cascella et al (2015) PLoSONE 10, e0121642 for better description of inducible and constitutive forms of HSP and the issues of classification.
L496-500: does not really say anything concrete.
L501 onwards: need better comparison to temperate and tropical sponge data and include the relevant experimental regimes used for them to give context.
L532: reference to “slight error”: this can easily be checked by manual annotation and needs validation before you start to describe significance.
L561: careful on references to GSTs: big gene family and function depends on orthology assignments.
L572: Is this whole sequence, or just match to functional domains? Please check as this can make a big difference.
L611: more on GPCRs would have been useful
L618-619: “all or nothing” response: the animals could be in a resistance phase, as the acute shock was for such a short period.
L639: comment on “little wiggle room” difficult to justify given experimental design. These data document an acute response only.

·

Basic reporting

The authors do a great job of providing extensive raw data files as well as tables and figures that describe particular aspects of the data set. Figure 6, however, needs to be enlarged with increased quality. In its current form, it is not possible to read the alignment, phylogeny, or amino acid identities. Also, for Table 4, it is unclear from the text how these genes were identified. Was a specific threshold change of expression used? It might help to include relative up and down regulation levels in the table.

Experimental design

The authors attempted to produce a transcriptomic data set with two technical replicates per treatment (control, 3oC, and 5oC). Their transcriptomes were of high quality and their assembly, testing for completeness, and annotation were well executed. It is unfortunate that one of their control replicates turned out to be a sample from a cryptic related species, but good that this was determined before moving forward with their transcriptomic analysis. This did mean, however, that for the control treatment, only one sample was sequenced. The authors are clear about this aspect of their experimental design and I don’t think that it invalidates their subsequent analyses, but I do think that more care should be taken in the interpretation of the data analyses (see below).

I am left wondering whether or not any other observations about the sponges in each treatment were recorded? Did the authors note what happens when sponges are left at those temperatures for longer periods of time – how quickly do they die compared to the control? Lines 569-571 states that “It seems likely that these sponges are under severe oxidative stress, and the expression of these genes reflects this.” Are there other metrics from the experiment that were measured or observed?

Validity of the findings

RSEM was used to quantify gene abundances and this is a preferred method when a reference genome is not in hand. Thus, even with the one control sample, the software can quantify transcripts in reference to the de novo transcriptome assembly. However, the lack of replication in the control treatment precludes robust statistical analysis. For example, in lines 341-345 the authors discuss “significant” changes in gene expression. More occurrences of “significance” appear in lines 364, 379, 380, 385, 387, 398, 399, 405, 407, and elsewhere. These sections should be clarified because it seems to me that statistical significance cannot be inferred when comparing the control (with a sample size of one replicate) to the 3oC or 5oC treatments. If the authors are not referring to statistical significance, then please define what is meant when the word “significant” is used.

The authors make good effort to connect their data to outcomes from other studies regarding thermal stress in sponges and other invertebrates. Further, there are interesting patterns of gene expression changes observed between the control sponge and the high temperature treatments. I do not agree, however, that the claim stated in the abstract and in the manuscript text, that the authors have demonstrated or studied how the sponges “adjust to environmental insult” is valid. Unless other metrics were measured that are not reported here, it seems to me that this study shows how gene expression patterns change when sponges are thermally stressed for four hours. Perhaps this could be referred to as a short-term thermal response of the sponges, but not an adjustment or “capacity to respond to thermal stress.” Further, it is interesting that the changes in gene expression between the 3oC and 5oC treatments were minimal (at least compared to the control). This is an important observation, but I’m not sure that it can be said that the data suggests “that moderate rises in temperature could cause stress at the limits of this organism’s capacity.” I don’t see how this was tested. I suggest revising this language to reduce the speculation (or state more clearly that this is speculation). Can you really say that changes in gene expression patterns (or lack thereof) are directly the result of cold-adapted sponges having limited range of tolerance? This may be a correlation, but causation has not been demonstrated.

Along these lines, the authors may want to modify their level of speculation regarding the role of particular gene products or pathways in the thermal response. Given that there are no functional studies related to the gene expression changes observed here, the most that can be said is that the data suggests particular genes may be involved in “thermal response.” For example, see lines 469-470, 479-482, 540-541, etc. I might suggest changing “likely” to “possible” and “confirms” to “suggests” to more accurately describe the data.

In reference to lines 632-635, I agree that the authors “generated an excellent reference transcriptome from a small number of replicate samples exposed to acute thermal stress,” and that they have “begun to identify the molecular responses” to short-term increased temperatures, but it is a stretch to say that these are the molecular responses “that these species will use to adjust to warmer ambient temperatures.” This part is quite speculative and not tested in this initial study. Please consider revising this sentence as well as 638-639. Again, without further testing, I’m not sure how this conclusion can be made.

Additional comments

This study is of clear importance and meaningful data sets were produced. The evaluation of gene expression changes was thorough. My main concern is regarding how the data are described in some places (e.g., whether or not “significance” should be used), and that some of the conclusions about the data are too speculative or have not been directly demonstrated by the study. If these issues can be revised, the manuscript will be improved.

---

## Round 0.2 · accepted · Accept

Great job addressing the reviewers' comments. I appreciate how difficult this work is and hope that future sampling can help add more robustness to these initial results.

·

Basic reporting

The paper is well-written and pleasant to read. Sufficient background on the study has been provided. Figures and tables are clear and supplementary data is available.

Experimental design

Research question is well defined and methods are sufficiently detailed.

Validity of the findings

Although general concerns on replication remain, the authors have done a great job of incorporating the limitations of their study (e.g. short-term exposure and limited replicates) into the results and discussion sections. They have also sufficiently justified their choice of parameters for certain methods (e.g. use of refseq database and e-value thresholds for annotation). The conclusions of the paper are well-stated with limitations clearly indicated, as necessary.

The comparison of differentially expressed genes in Isodictya versus other sponge species is interesting as it confirms some common temperature-response pathways. This is a valuable addition to our current understanding on the temperature-induced responses of diverse sponge species.

Additional comments

The authors have sufficiently addressed reviewer comments and suggestions. They have managed to clarify the limitations/challenges of the study without minimizing the value of their findings. They also present recommendations for future studies that are certainly worth pursuing.